



# North Atlantic Oscillation controls on oxygen and hydrogen isotope gradients in winter precipitation across Europe; implications for palaeoclimate studies

Michael Deininger[1], Martin Werner[2], Frank McDermott[1,3]

[1]UCD School of Earth Sciences, University College Dublin, Belfield, Dublin 4, Ireland
[2]Alfred Wegener Institute, Helmholtz Centre for Polar and Marine Research, Bussestraße 24, 27570 Bremerhaven, Germany
[3]UCD Earth Institute, University College Dublin, Belfield, Dublin 4, Ireland

*Correspondence to*: Michael Deininger (michael.deininger@ucd.ie)

**Abstract.** Winter (October to March) precipitation $\delta^{18}O_P$ and $\delta D_P$ values in central Europe correlate with the winter NAO index (wNAOi), but the causal mechanisms remain poorly understood. Here we analyse the relationships between precipitation-weighted $\delta^{18}O_P$ and $\delta D_P$ datasets ($\delta^{18}O_{pw}$ and $\delta D_{pw}$) from European GNIP and ANIP stations and the wNAOi, with a focus on isotope gradients. We demonstrate that longitudinal $\delta^{18}O_{pw}$ and $\delta D_{pw}$ gradients across Europe (continental effect) depend on the wNAOi state, with steeper gradients associated with more negative wNAOi states. Changing gradients reflect a combination of air temperature and variable amounts of precipitable water as a function of the wNAOi. The relationships between the wNAOi, $\delta^{18}O_{pw}$ and $\delta D_{pw}$ can provide additional information from palaeoclimate archives such as European speleothems that primarily record winter $\delta^{18}O_{pw}$. Comparisons between present-day and past European longitudinal $\delta^{18}O$ gradients inferred from Holocene speleothems suggest that negative wNAO modes dominated the early Holocene, but positive wNAO modes were more common in the late Holocene.

## 1 Introduction

Stable oxygen and hydrogen isotopes ($\delta^{18}O$, $\delta D$; relative to Vienna Standard Mean Ocean Water) in precipitation have been analyzed with great effort since the 1950s (Dansgaard, 1954;Craig, 1961;Craig and Gordon, 1965) to decipher environmental processes that control their variations and to facilitate their use in palaeoclimatology (Dansgaard, 1964;Gat, 1996;Rozanski et al., 1992;Aggarwal et al., 2012). The Global Network of Isotopes in Precipitation (GNIP) program initiated in 1958 by the International Atomic Energy Agency (IAEA) and the World Meteorological Organization (WMO) has operated since 1961, and has made great progress towards these aims (e.g. Dansgaard, 1964;Rozanski et al., 1992). For the European sector of the GNIP program, 438 stations have operated, with the oldest records from the GNIP stations at Valentia Observatory in Ireland (since 1958), and at Hohe Warte (Vienna, Austria) since 1960. The processes that drive $\delta^{18}O$ values in precipitation ($\delta^{18}O_P$) at these stations have been discussed extensively in the literature, focusing mainly on air temperature, altitude and the continental effect (Rozanski et al., 1982;Stumpp et al., 2014;Schürch et al., 2003;Lykoudis and Argiriou, 2011;Fischer and



Baldini, 2011). In recent years, several studies have investigated the relationship between the North Atlantic Oscillation (NAO) – the major atmospheric mode of European winter climate, (Hurrell et al., 2003;Hurrell and VanLoon, 1997;Hurrell, 1995) – and $\delta^{18}O_P$ at specific stations (Baldini et al., 2008;Langebroek et al., 2011;Field, 2010). More recently, the effect of

the East Atlantic pattern, the second major mode of atmospheric pressure variability in the North Atlantic region (Barnston and Livezey, 1987) on the relationship between the winter NAO index (wNAOi) and the $\delta^{18}O_P$ in Europe has been examined (Comas-Bru et al., 2016). A complementary approach, developed here, is to document and seek to understand the *longitudinal gradients* in winter rainfall $\delta^{18}O_P$ across Europe, and to link these to different states of the NAO.

Here we present a detailed station-based evaluation of the effects of the wNAOi on precipitation-weighted $\delta^{18}O_P$ and $\delta D_P$ values ($\delta^{18}O_{pw}$ and $\delta D_{pw}$) during the winter season for continental Europe and the Alps. We demonstrate that the well-documented longitudinal gradient in $\delta^{18}O_{pw}$ and $\delta D_{pw}$ across the European continent (often referred to as the 'continental effect') depends on the state of the winter NAO index (wNAOi), through a combination of air temperature and atmospheric precipitation history effects. The downstream effects of this wNAOi–linked variability can also be detected in the $\delta^{18}O_{pw}$ and

$\delta D_{pw}$ values at stations north of the Alpine divide. We also make use of the $\delta^{18}O_{pw}$ and $\delta D_{pw}$ outputs from an isotope-enabled General Circulation Model ECHAM5-wiso (Werner et al., 2011;Langebroek et al., 2011) to better evaluate the NAO-dependence on isotope longitudinal gradients. The model data evaluation shows that, first, the observed NAO-dependence of the $\delta^{18}O_{pw}$ and $\delta D_{pw}$ and longitudinal gradients are reproduced by the ECHAM5-wiso outputs, and second that the variability of the ECHAM5-wiso longitudinal gradients are in the same range as the longitudinal gradients observed in the station data.

This study improves our understanding of the significance of inferred past changes in $\delta^{18}O_{pw}$ or $\delta D_{pw}$ gradients across Europe, for example during the course of the Holocene (e.g. McDermott et al., 2011) or the last 35 ka (e.g. Rozanski, 1985), by linking modern $\delta^{18}O_{pw}$ and $\delta D_{pw}$ gradients, respectively, to different states of the wNAO and/or changes in the European continental temperature gradient and atmospheric precipitation histories.

## 2 Data and Methods

### 2.1 Station-based data

Monthly winter (October to March) $\delta^{18}O_P$ and $\delta D_P$, temperature and precipitation data from 37 European stations were used to investigate the dependence of the precipitation-weighted winter $\delta^{18}O_P$ and $\delta D_P$ values ($\delta^{18}O_{pw}$ and $\delta D_{pw}$) on the wNAOi (Figure 1).

Twenty-eight of these stations are part of the GNIP network and six are part of the Austrian Network of Isotopes in Precipitation (ANIP). The data from the ANIP stations originate from Humer (1995) and the ANIP homepage (http://wisa.bmlfuw.gv.at/daten.html). The GNIP stations were selected based on recent findings, showing that the un-weighted winter (December to March) $\delta^{18}O_P$ and $\delta D_P$ values at these stations are significantly correlated with the wNAOi



(Baldini et al., 2008). The ANIP stations were selected for their long records starting in the early 1970s and their location,
which ensures that there are sufficient $\delta^{18}O_P$ and $\delta D_P$ measurements in the Alps for the purposes of this study. For our
analysis, the wNAOi is the average winter value calculated from the monthly PC-based (Principal Component) NAOi values
from December to March (Hurrell, 1995) downloaded from the Climate Data Guide[1]. For every station, the precipitation-
weighted $\delta^{18}O_P$ and $\delta D_P$ values ($\delta^{18}O_{pw}$ and $\delta D_{pw}$) were calculated for each winter, using the monthly $\delta^{18}O_P$ and $\delta D_P$ values,
weighted by the monthly precipitation amounts, for the period October to March (6 month) and December to February (3
month). The studied periods are chosen on the one hand because during December to February the NAO exerts its strongest
influence on the European winter climate (e.g. Hurrell, 1995) and we expect to observe the strongest influence on $\delta^{18}O_P$ and
$\delta D_P$ in this 3-month period and on the other hand because the main infiltration period for precipitation in Central Europe is
from October to March and is, therefore, the period of interest for palaeoclimate archives recording the infiltrated $\delta^{18}O_P$ and
$\delta D_P$ in this 6-month interval (e.g. speleothems). For further analysis, the winter $\delta^{18}O_{pw}$ and $\delta D_{pw}$ values (6 month and 3
month) of every station are grouped into six classes depending on the wNAOi. Thus, the wNAOi was sub-divided into six
classes (bins): (i) $1.6 \leq$ wNAOi; (ii) $0.8 \leq$ wNAOi $< 1.6$; (iii) $0 \leq$ wNAOi $< 0.8$; (iv) $-0.8 \leq$ wNAOi $< 0$; (v) $-1.6 \leq$ wNAOi $<$
$-0.8$; (vi) wNAOi $< -1.6$, resulting in six compilations of $\delta^{18}O_{pw}$ and $\delta D_{pw}$ values for every station. For every station, the
median of every compilation was calculated. An example is given in Figure 2, which shows how the data were processed,
resulting in six median $\delta^{18}O_{pw}$ and $\delta D_{pw}$ values for the Garmisch-Partenkirchen station (#24 in Figure 1).

For the analysis of the processed $\delta^{18}O_{pw}$ and $\delta D_{pw}$ values, the stations were also subdivided into continental stations, which
includes all stations with an altitude $\leq 350$m and having no (or negligible) Mediterranean influence (13 stations); and stations
with an altitude $>350$m. The latter stations were divided into two groups: continental stations with an altitude higher than
350m (3 stations) and Alpine stations (17 stations). Furthermore, four Mediterranean GNIP stations south of the Alps were
analysed to better validate the $\delta^{18}O_{pw}$ and $\delta D_{pw}$ values of the Alpine regions (Figure 1).

## 2.2 Isotope modelling analysis (ECHAM5-wiso)

To independently evaluate possible mechanisms that determine the longitudinal $\delta^{18}O_{pw}$ and $\delta D_{pw}$ gradients, the $\delta^{18}O_{pw}$ and
$\delta D_{pw}$ data from the isotope-enabled General Circulation Model ECHAM5-wiso (Langebroek et al., 2011;Werner et al., 2011)
were analysed for the period 1960-2010 AD. The analysed ECHAM5-wiso model output has a spatial resolution of
1.9°x1.9°. (Details on the ECHAM5-wiso model and the ECHAM5-wiso data NAO-dependence are given by Werner et al.
(2011) and Langebroek et al. (2011) respectively). To compare the ECHAM5-wiso data with the station-based data, only the
model output for those grid cells where a continental station is located (Figure 1) were analysed. For these selected grid cells,
the 3-month (December-February) and 6-month (October-March) $\delta^{18}O_{pw}$ and $\delta D_{pw}$ values were calculated. These model

---

[1] The Climate Data Guide: Principal Component based North Atlantic Oscillation (NAO) Index:
https://climatedataguide.ucar.edu/climate-data/hurrell-north-atlantic-oscillation-nao-index-pc-based.





winter values were grouped (as described in Section 2.1) using the ECHAM5-wiso winter NAO index. The monthly
ECHAM5-wiso NAO index is calculated by Principal Component Analysis (PCA) from the model monthly sea level
pressure field in the sector from 40 °W to 40 °E and 20 °N to 80 °N (see Langebroek et al., 2011 for details on the
calculation of the ECHAM5-wiso NAO index). The ECHAM5-wiso winter NAO index is the mean NAO index for the
months December to March. For every grid cell, this approach results in six median $\delta^{18}O_{pw}$ and $\delta D_{pw}$ (one median value for
every NAO class) of which the model longitudinal $\delta^{18}O_{pw}$ and $\delta D_{pw}$ gradient is calculated.

## 100   3 Results

### 3.1 Results of the longitudinal $\delta^{18}O_{pw}$ and $\delta D_{pw}$ gradients (continental stations; ECHAM5-wiso)

To study the relationship between the wNAOi classes and the longitudinal $\delta^{18}O_{pw}$ and $\delta D_{pw}$ gradients during the winter
season, $\delta^{18}O_{pw}$, $\delta D_{pw}$ datasets of a total of 13 'continental' GNIP stations are analysed. The most westerly station is at -10.25
°E (Valentia Observatory; Ireland) and the easternmost station is at 19.85 °E (Krakow, Poland) (Figure 1). These continental
stations show a positive sensitivity of $\delta^{18}O_{pw}$ and $\delta D_{pw}$ to the wNAOi (Figure 3), i.e., a positive linear relationship with the
wNAOi for the investigated continental station datasets, resulting in more positive median $\delta^{18}O_{pw}$ and $\delta D_{pw}$ values for higher
wNAOi classes, as in Figure 2 for Garmisch-Partenkirchen. An exception to this observation are the GNIP stations at the
Valentia Observatory (Station 1) where the $\delta^{18}O_{pw}$ and $\delta D_{pw}$ values do not show a clear linear relationship compared to the
other continental stations for either the 3-month (Figure 3a and 3c) or 6-month (Figure 3b and 3d) winter periods. Similar
behaviour is seen in the data for the Wallingford GNIP station (Station 2) for the 6-month winter period (Figure 3b and 3d).
It is notable, that for most stations, the sensitivity of $\delta^{18}O_{pw}$ and $\delta D_{pw}$ to the wNAOi is higher for the 3-month winter period
than for the 6-month winter period (Figure 3). The median sensitivity for $\delta^{18}O_{pw}$ is 0.87 (3 month) and 0.57 (6 month)
‰/wNAOi unit, and for $\delta D_{pw}$ it is 6.87 (3 month) and 4.44 (6 month) ‰/wNAOi unit. This likely reflects the fact that the
mode of the wNAOi (December to March) exerts the strongest influence on European winter meteorology (e.g. Hurrell,
1995;Hurrell et al., 2003), and so on $\delta^{18}O_{pw}$ and $\delta D_{pw}$ (see discussion in Section 4.1).

To investigate the dependence of the longitudinal $\delta^{18}O_{pw}$ and $\delta D_{pw}$ gradients from these 13 continental stations on the
wNAOi classes, the gradient slopes were calculated for each of the six wNAOi classes. For all winter month periods the
slope is negative (towards the east) indicating that the $\delta^{18}O_{pw}$ and $\delta D_{pw}$ values become more negative towards the east.
Furthermore, the slopes of the longitudinal $\delta^{18}O_{pw}$ and $\delta D_{pw}$ gradients of the continental stations becomes steeper for winters
with more negative wNAOi classes (Figure 4). This means that $^{18}O$ and $^2H$ are more efficiently removed from the
atmospheric moisture with distance from the western margin, resulting in more strongly depleted $\delta^{18}O_{pw}$ and $\delta D_{pw}$ values
along the longitudinal transect during more negative NAO winters compared with more positive NAO winters. Furthermore,
the slope is steeper for the 3-month (DJF) averages compared with that for the 6-winter month averages.






Comparison of the longitudinal $\delta^{18}O_{pw}$ and $\delta D_{pw}$ gradients derived from the ECHAM5-wiso with those from the station-based data show that slopes from the ECHAM5-wiso data reproduce the observed station-based slopes quite well (Figure 4). Only the slopes determined for the most negative wNAOi class of the 6-month winter period level off from the empirically determined slopes (Figure 4a and 4c). Overall, however, there is good agreement between the ECHAM5-wiso isotope gradients and those derived from the observational datasets is very good.

### 3.2 Alpine stations and other stations with an altitude > 350m

Data from European stations with an altitude greater than 350m are separated into two groups: Alpine stations and non-Alpine stations. The non-Alpine stations include Wasserkuppe-Rhoen (#20), Hof-Hohensaas (#28) and Regensburg (#29), all located in Germany (Figure 1). These show similar $\delta^{18}O_{pw}$ and $\delta D_{pw}$ sensitivity to the wNAOi as the other continental stations (Figure 3). The Alpine stations include a total of 17 stations that are well distributed over the Alps (Figure 1). These data allow an evaluation of precipitation $\delta^{18}O_{pw}$ and $\delta D_{pw}$ values and patterns in the entire Alpine region as a function of the wNAOi. The sensitivity analysis of $\delta^{18}O_{pw}$ and $\delta D_{pw}$ of the Alpine stations reveals a somewhat more complex relationship to the wNAOi. For the 3-month winter period, the $\delta^{18}O_{pw}$ data from the Thonon-Les-Bains (14), Längenfeld (21), Obergurgl (23), Böckstein (30), St. Peter (31), Villacher Alpe (32), Graz University stations (33) have a weak or absent relationship to the wNAOi, while the $\delta^{18}O_{pw}$ and $\delta D_{pw}$ data from the other Alpine stations have a similar sensitivity to the wNAOi as the continental stations (Figure 3a). The $\delta D_{pw}$ data show similar results as $\delta^{18}O_{pw}$, but there are different results for some other stations. For Thonon-Les-Bains (14) the relationship between $\delta D_{pw}$ and the wNAOi is stronger compared to $\delta^{18}O_{pw}$ whereas the relationship for Grimsel is weaker (18) (Figure 3c). For the 6-month winter period, the $\delta^{18}O_{pw}$ sensitivity to the wNAOi for most of the Alpine stations is comparably strong to that of the continental stations. Only the stations at Thonon-Les-Bains (14), Längenfeld (21), Böckstein (30), St. Peter (31), Villacher Alpe (32), Graz University (33) show a weak relationship to the wNAOi (like for the 3-month winter period). The Obergurgl station (23) has a stronger relationship to the wNAOi – compared with the 3-month winter period – but has a smaller average $\delta^{18}O_{pw}$ sensitivity compared with the other Alpine stations. The highest $\delta^{18}O_{pw}$ sensitivity can be found for Hohenpeißenberg (22) and Regensburg (29) and is 0.77 ‰/wNAOi unit (Figure 3b). The $\delta D_{pw}$ sensitivity of the Alpine stations for the 6-month winter period shows a similar relationship as for the 3-month winter period. For the Bern (15), Guttannen (17), Obergurgl (23) and Villacher Alpe (32) stations, however, the conclusion is different compared with the 3-month winter period. The relationship of $\delta D_{pw}$ to the wNAOi is weaker for Bern (17) whereas it is stronger for the other stations. The highest sensitivity is observed for the Thonon-Les-Bains (14) and Meiringen (16) stations and is 7.42 and 8.84 ‰/wNAOi unit, respectively (Figure 3d).





## 4 Discussion

### 4.1 Longitudinal $\delta^{18}O_{pw}$ and $\delta D_{pw}$ gradients

To explain the change in the slopes of the longitudinal winter $\delta^{18}O_{pw}$ and $\delta D_{pw}$ gradients (Figure 4), the measured temperatures and the amount of precipitation at the GNIP stations and from the relevant ECHAM5-wiso model grid cells were evaluated. Furthermore, the total precipitable water was analysed for the ECHAM5-wiso dataset. These variables were also grouped according to wNAOi class of their respective winters, and the median of temperature and precipitation were calculated for each wNAOi class (Figures 5 and 6).

The GNIP station based continental temperature and precipitation gradients for the different wNAOi classes show that the slopes of the longitudinal winter temperature gradients become steeper for lower wNAOi values and are always negative (i.e. average winter temperatures are always lower in the east), but no equivalent relationship is observed for the slopes of the winter precipitation gradients (Figure 5). The temperature relationships suggests that while the winter air temperature clearly becomes colder from west to east, a higher average air temperature gradient between western and eastern Europe occurs in more negative wNAOi winters. Furthermore, the intercept of the linear regression is progressively smaller for more negative wNAOi modes, suggesting general cooler conditions in central Europe during negative wNAOi modes. This is consistent with the general relationship between the wNAOi and winter air temperatures for central Europe (e.g. Hurrell, 1995;Comas-Bru and McDermott, 2013). Curiously, for the most negative wNAOi class, the slope of the temperature gradient does not follow the general trend and has a lower value, comparable to more positive winter wNAOi modes, suggesting a smaller temperature difference between western and eastern Europe under these conditions. The reason for this change is unclear and reflects possibly a relationship of the cyclone variability to the NAO (Gulev et al., 2001), which may increase the frequency of incursions of cold easterly winds into western Europe during very negative wNAO modes.

Similar temperature-NAO relationships are observed for the ECHAM5-wiso simulations, with increasingly steeper model air temperature gradients and a smaller temperature intercept for lower wNAO indices (Figure 6a and 6b). However, in comparison with the slopes derived from the GNIP datasets, the slopes derived from the ECHAM5-wiso simulations suggest steeper temperature gradients (Figure 6a). The intercept of the temperature regression derived from the ECHAM5-wiso simulations is similar to that from the observational datasets (Figure 6b). Therefore, the longitudinal temperature gradients of the ECHAM5-wiso simulations suggest comparable temperatures at 0°E (the temperature intercept), but much cooler temperatures in eastern Europe (a steeper temperature slope) compared with the observed air temperatures in the GNIP datasets.

The observed (GNIP datasets) and simulated (ECHAM5-wiso) temperature slopes can be used to calculate the expected air temperature-driven difference in $\delta^{18}O_{pw}$ and $\delta D_{pw}$ between the western- and eastern-most GNIP stations using theoretical temperature sensitivities for $\delta^{18}O_P$ and $\delta D_P$ (e.g. from Dansgaard, 1964). The theoretical changes in $\delta^{18}O_{pw}$ and $\delta D_{pw}$ between



the western- and eastern-most GNIP stations (the longitudinal difference between the Valentia (Observatory) and Krakow (Wola Justowska) is 30.1°) were calculated as follows. First, the temperature difference between these two stations was calculated from the temperature slope for the different wNAOi classes. For example, for the highest wNAOi class, the slope

of the observed temperature gradient is -0.19 K/°E and -0.26 K/°E for the 6-month and 3-month winter period, resulting in a temperature difference of 5.86 K and 7.78 K, respectively. (By comparison, in the ECHAM5-wiso output, the temperature difference is somewhat greater at 7.73 K and 9.64 K for the highest wNAO class for the 6-month and 3-month winter periods, respectively.) Hence, the average winter temperature at Valentia Observatory from October to March (December to February) is about 5.86 K (7.78 K) warmer compared to Krakow Wola Justowska for the highest wNAO class. The effect of

this eastward temperature decrease can be converted into an expected $\delta^{18}O_{pw}$ and $\delta D_{pw}$ difference between the two stations. For this estimate we apply an approximate estimation of the sensitivity of $\delta^{18}O_P$ and $\delta D_P$ on temperature changes based on theoretically derived values by Dansgaard (1964), assuming a Rayleigh-type moist adiabatic condensation (vapour-liquid) process. For the sensitivity, the average value for a cooling from an initial temperature of 0°C to -20°C is used, which is 0.64 ‰/K for $\delta^{18}O_P$ and 5.6 ‰/K for $\delta D_P$ (Dansgaard, 1964). Hence, for the highest wNAO class, the observed temperature

difference would cause a calculated difference of 3.75 ‰ (6-month) and 4.98 ‰ (3-month) for $\delta^{18}O_{pw}$. $\delta^{18}O_{pw}$ values are therefore expected to be 3.75 ‰ and 4.98 ‰ lighter at the easternmost station compared to the westernmost station (for the 3 and 6 winter month averages respectively).

These theoretical calculated values are now compared with the observed differences derived from the slopes of the linear regression of the GNIP $\delta^{18}O_{pw}$ datasets (Figure 4). For the highest wNAO class, the slope is -0.16 ‰/°E and 0.20

‰/°E for the 6-month and 3-month winter period. This results in an observed difference of 4.78 ‰ (6-month) and 6.02 ‰ (3-month). Importantly, the observed differences (longitudinal gradients) are much larger than those calculated using the air-temperature driven Dansgaard-type model described above. The results of these calculations for $\delta^{18}O_{pw}$ and $\delta D_{pw}$, based on the observed temperature slopes, and for all wNAO classes are listed in Table 1.

The most important result of this simple exercise is that the observed differences in $\delta^{18}O_{pw}$ and $\delta D_{pw}$ between the western- and eastern-most GNIP stations are larger than can be accounted for by an simple air temperature driven Rayleigh distillation model alone (Dansgaard, 1964). Repeating the calculations using the vapour-ice phase change (snow) instead results in calculated differences that are still too small to explain the observed differences in $\delta^{18}O_{pw}$ and $\delta D_{pw}$ between the western- and eastern-most stations (not shown). This strongly indicates that the observed longitudinal variations in winter air

temperatures alone simply are insufficient to account for the observed difference between $\delta^{18}O_{pw}$ and $\delta D_{pw}$ in western and eastern Europe, and that additional processes must be considered.

By contrast with the observational (GNIP) data discussed above, the ECHAM5-wiso simulated differences in $\delta^{18}O_{pw}$ and $\delta D_{pw}$ can largely be accounted for by model air-temperature differences alone. Only for the most negative winter NAO class does the expected temperature-driven west to east change in $\delta^{18}O_{pw}$ and $\delta D_{pw}$ exceed the ECHAM5-wiso

simulated differences derived from slope of the lowest wNAOi class. If the slope of the ECHAM5-wiso longitudinal $\delta^{18}O_{pw}$





and $\delta D_{pw}$ gradients were driven by winter air temperature gradients alone, the average temperature sensitivity for the remaining five wNAOi classes for the 6-month winter period for $\delta^{18}O_{pw}$ and $\delta D_{pw}$ calculated from the ECHAM5-wiso simulations varies between 0.59 and 0.63 ‰/K for $\delta^{18}O_{pw}$ and 4.44 and 4.87 ‰/K for $\delta D_{pw}$. For the 3-month winter period, equivalent temperature sensitivities from the ECHAM5-wiso simulations range between 0.51 and 0.58 ‰/K for $\delta^{18}O_{pw}$ and

4.09 and 4.65 ‰/K for $\delta D_{pw}$ and are smaller than for the 6-month winter period. Hence, the temperature sensitivities for $\delta^{18}O_{pw}$ and $\delta D_{pw}$ derived from the ECHAM5-wiso are apparently somewhat lower compared to the theoretically estimated ones based on the approach of Dansgaard (1964). One explanation for the larger temperature sensitivities derived by the Dansgaard (1964) approach may be that the initial temperature at which the moisture condensation begins in the atmosphere is actually greater than 0°C as assumed in our theoretical calculations. Higher condensation temperatures are reasonable

considering that the temperature intercepts derived from the observed temperatures vary between c. 3 °C and 7 °C (Figure 5b). Overall, however, these results supports the conclusion above that the winter air temperature effect on the longitudinal winter $\delta^{18}O_{pw}$ and $\delta D_{pw}$ gradients is insufficient to explain the observed difference between the western and eastern GNIP stations. It is intriguing that the observed (GNIP datasets) and simulated (ECHAM5-wiso simulations) temperature slopes differ (Figure 6), while the slopes for longitudinal $\delta^{18}O_{pw}$ and $\delta D_{pw}$ gradients are apparently similar (Figure 4).

Notwithstanding these differences in the observed and simulated temperature gradients, our calculations suggest that the temperature gradient has the largest influence on the longitudinal isotope gradient. This is indicated by the deviation between the calculated and the observed difference between the western- and eastern-most GNIP stations (Table 1). Furthermore, the deviation is larger for more negative wNAO classes suggesting that the additional processes become relatively more important for lower wNAOi values.


Rozanski et al. (1982) pointed out that the precipitation history of air masses is an important control on observed longitudinal $\delta D_P$ gradients. Precipitation history can be expressed as a numerical value f (fraction of remaining moisture). Hence, f depends on the balance between the amount of precipitation (P) that has already occurred along the longitudinal gradient and the initial amount of precipitable water ($Q_0$). However, only weak relationships are found between the precipitation gradients

calculated from the GNIP station precipitation datasets and the wNAOi classes, suggesting that rainfall gradients between western and eastern Europe are fairly constant, within the range of uncertainty (Figure 5c and 5d). Furthermore, the intercept of the linear regression of the precipitation data shows no dependence with the class of the wNAOi. This is consistent with the findings of Baldini et al. (2008) who showed that precipitation data from continental GNIP stations have no systematic correlation to the wNAOi (opposite to temperature). As discussed below, this points to differences in the initial amounts of

precipitable water ($Q_0$) as a possible control on isotope gradient-wNAOi relationships (Figure 4), over and above those attributable to air temperature gradients alone.

By contrast, in the ECHAM5-wiso simulations, a clear relationship exists between the slope and the intercept of the longitudinal precipitation gradient (Figure 6c and 6d). The slopes, as well as the intercepts of the precipitation gradients are smaller for lower wNAOi classes, suggesting that the west-east difference of the amount of precipitation decreases





(increases) for lower (higher) wNAOi classes in the model. Furthermore, the decreasing (increasing) intercept of the precipitation gradient for lower (higher) wNAOi classes suggests a lower (higher) amount of rainfall in the western grid cells. Analysis of the total amount of precipitation for all grid cells reveals lower values for the total amount of rainfall in the selected grid cells (i.e. the sum of precipitation from all grid cells) for lower wNAOi classes. For the 6-month winter period, for example, the median of the total precipitation decreases from 1091 mm for the highest wNAOi class to 852.5 mm for the

lowest wNAOi class. In the case of the 3-month winter period, the median of the total precipitation decreases from 1270 mm to 800 mm from the highest to the lowest wNAOi class. For the lowest wNAOi class, the total amount of precipitation of all grid cells is about 78 % and 63 % of that from the highest wNAOi class for the 6-month and 3-month winter periods, respectively. Thus, the ECHAM5-wiso output is not consistent with the observed data for precipitation at the GNIP stations, which show only a very weak relationship with the wNAOi class ($R^2$=0.07). Indeed, in the observational data, the total

amount of precipitation for the lowest wNAOi class is actually marginally higher compared with that for the highest wNAOi class (c. 762 mm compared to 866 mm for the 6-month winter period, and c. 797 mm compared to 830 mm for the 3-month winter period). This comparison indicates that the precipitation history inferred from the observational datasets and the ECHAM5-wiso simulations are quite different. The effect of these differences on the longitudinal $\delta^{18}O_{pw}$ and $\delta D_{pw}$ gradients are discussed below.


To better constrain the governing physical mechanisms of the precipitation history f and their first-order effects on $\delta^{18}O_{pw}$ and $\delta D_{pw}$, a simple one-box Rayleigh-type model for the atmosphere was used (Dansgaard, 1964;Eriksson, 1965):

$$^xR = \ ^xR_0 \cdot f^{\,x\alpha-1} = \ ^xR_0 \left(\frac{Q}{Q_0}\right)^{x\alpha-1} = \ ^xR_0 \left(1 - \frac{P}{Q_0}\right)^{x\alpha-1}. \tag{1}$$


Equation (1) is the classic Rayleigh distillation model (Rayleigh, 1902;see also Mook, 2006 for detail) that describes the evolution of an isotope ratio R (the subscript x is a place holder for x = 2 ($^2$H) or 18 ($^{18}$O), i.e., for $^{18}R=^{18}O/^{16}O$ and $^2R=^2H/^1H$) as a function of the precipitation history f. This is a function of the amount of initial ($Q_0$) and remaining (Q) precipitable water in the atmosphere; Q is therefore the amount of precipitable water after a specific amount of precipitation

P ($Q_0$-Q=P) has formed. $R_0$ describes the initial isotope ratio. $\alpha$ is the equilibrium liquid-water isotope fractionation factor that depends only on temperature; the subscript x denotes, as above, the related isotope system. Because the slope of the longitudinal precipitation gradient does not change systematically with the wNAOi class, P does not change with the class of the wNAOi, and has, therefore, a negligible effect on $\delta^{18}O_{pw}$ and $\delta D_{pw}$. Hence, the slope of the longitudinal gradient of $\delta^{18}O_{pw}$ and $\delta D_{pw}$ in central Europe is driven only by the temperature dependent isotope fractionation factor $\alpha$, and by the

initial amount of precipitable water $Q_0$ in the atmosphere. Although the classic Rayleigh-type model, adopted here is unable to fully capture all atmospheric processes (e.g. mixing of atmospheric moisture with different isotope signatures and/or origins), it is nonetheless a useful first approximation to explain the deviations between the calculated (temperature effect)





and observed differences in $\delta^{18}O_{pw}$ and $\delta D_{pw}$ between the western- and eastern-most stations. Therefore, our observations of the dependence of the longitudinal $\delta^{18}O_{pw}$ and $\delta D_{pw}$ gradient on the class of the wNAOi can be explained if additionally the

amount of precipitable water over central Europe is lower for more negative wNAOi values. As demonstrated previously in a comparison of very positive (wNAOi>1) and negative (wNAOi<-1) wNAOi values, the maximum amount of precipitable water in the atmosphere is shifted southward for very negative wNAOi (Trigo et al., 2002). This shift in the amount of precipitable water is associated with changing air temperature patterns, and smaller amounts of precipitable water are associated with cooler continental air temperatures. Accordingly, our simple model shows that the atmosphere in central

Europe contains less (more) atmospheric moisture during more negative (positive) wNAOi states, and this is independently confirmed by analysis of the amount of precipitable water in the atmosphere from the ECHAM5-wiso dataset (see below). Hence, atmospheric moisture $\delta^{18}O$ and $\delta D$ values are likely to be more sensitive to the rainout history during more negative wNAOi modes, because f must change at a higher rate if $Q_0$ is smaller and P is held constant (Eq. 1). This effect is also confirmed by a multi-box exercise that assumes a Rayleigh-type condensation process, mimicking the longitudinal gradients.

This multi-box exercise shows that $\delta^{18}O_P$ becomes progressively depleted in dependence on $Q_0$ from west to east across Europe, with steeper gradients (higher slopes) for smaller $Q_0$ (Figure S1).

The wNAOi-precipitable water relationship described by Trigo et al. (2002) is evident in the ECHAM5-wiso simulations. Precipitable water shows a positive correlation with wNAOi values in central and northern Europe, and a negative

correlation over the Mediterranean (including Iberia, the Balkans and Turkey) (Figure S2). This indicates that during positive wNAO modes, the amount of precipitable water increases over central Europe and decreases during negative wNAO modes. This finding is also confirmed by the amount of precipitable water of the analysed ECAHM5-wiso grid cells (Figure 6e and 6f). The results of the regression analysis of the longitudinal gradient of the modelled precipitable water shows that the slope of the precipitable water along the longitudinal gradient is independent of the wNAOi class (Figure 6e), while the intercept

decreases for smaller wNAOi classes (Figure 6f). Compared with the amount of precipitable water for the highest wNAOi class, the atmosphere contains only about 85.7 % for the lowest wNAOi class for the 6-month winter period for example. In the ECHAM5-wiso simulations, the total amount of precipitation (78 % for the 6-month winter period) decreases more along the longitudinal transect than does the amount of precipitable water. As a result, the precipitation history f becomes less sensitive to the rainout history along the longitudinal gradient for lower wNAOi classes. The differences between the

theoretically derived temperature sensitivity by Dansgaard (1964) and the temperature sensitivity derived by the ECHAM5-wiso simulations could thus be ascribed to the changing ratio of total precipitation and amount of precipitable water as seen in the ECHAM5-wiso simulations.

To summarise, the dependence of the *observed* longitudinal $\delta^{18}O_{pw}$ and $\delta D_{pw}$ gradient on the class of wNAOi in the winter

season results from two processes: (i) the changing continental temperature gradients via the temperature dependent isotope fraction during condensation, which exerts the strongest influence on the $\delta^{18}O_{pw}$ and $\delta D_{pw}$ gradient and (ii) the dependence of





the amount of precipitable water or in general on the precipitation history, over central Europe on the wNAOi mode, which becomes important for more negative wNAOi classes. The latter mechanism is in agreement with recent findings of Aggarwal et al. (2012) who showed that, generally, more negative $\delta^{18}O_P$ values are associated with lower moisture residence

times, where the moisture residence time is defined as the ratio between the amount of precipitable water and the precipitation (Aggarwal et al., 2012;Trenberth, 1998) and conclusions of (Rozanski et al., 1982) analysing summer and winter European $\delta D_P$ longitudinal gradients. For the *ECHAM5-wiso*, air temperature gradients are clearly the most important factor that controls the longitudinal $\delta^{18}O_{pw}$ and $\delta D_{pw}$ gradients, In *ECHAM5-wiso*, the precipitation history seems to be of relatively minor importance, because changes in the amount of precipitable water are mediated by changes in the simulated

amount of precipitation. The reason for the different strength of these two mechanisms (temperature gradient and precipitation history) on the longitudinal $\delta^{18}O_{pw}$ and $\delta D_{pw}$ gradients for the observed (GNIP) and simulated (ECHAM5-wiso) datasets remains unclear, suggesting that the ECHAM5-wiso simulations warrant further investigation.

### 4.2 Alpine stations

By comparison with the low-altitude stations, the Alpine stations reveal more complex wNAOi - $\delta^{18}O_{pw}$, $\delta D_{pw}$ patterns

(Figure 3). North of the Alpine divide, all stations show similar $\delta^{18}O_{pw}$ and $\delta D_{pw}$ - wNAOi class relationships as at the Garmisch-Partenkirchen GNIP station (Figure 2). Precipitation at these stations is depleted in $^{18}O$ and $^{2}H$ for more negative wNAOi winters. The only exceptions to this relationship are the $\delta^{18}O_{pw}$ datasets for Thonon-Les-Bains, whose $\delta^{18}O_{pw}$ datasets have only a weak relationship to the wNAOi. The relationship between the $\delta^{18}O_{pw}$ - $\delta D_{pw}$ values and wNAOi from stations at and south of the Alpine divide, including Grimsel (#18) (western Alps); Längenfeld (#21), Obergurgl (#23),

Patscherkofel (#27) (all central Alps); Böckstein (#30), St. Peter (#31), Villacher Alpe (#32) and Graz (#33) (eastern Alps) is more complex compared to the stations north of the Alpine divide. For the 3-month $\delta D_{pw}$ datasets only, the isotope data from Patscherkofel show a strong relationship to the wNAOi, while all other stations have weak or no relationships to the wNAOi (Figure 3c). In the 6-month dataset, only the southern-most stations (Längenfeld and St. Peter) show a weak relationship to the wNAOi. All other stations have a strong relationship to the wNAOi (Figure 3d). The $\delta D_{pw}$ dataset of the southern- and

eastern-most Alpine stations, Villacher Alp (32) and Graz (33), respectively, exhibit no relationships with the wNAOi for the 3-month datasets; for 6-month winter period only the $\delta D_{pw}$ dataset of Villacher Alpe has a relationship to the wNAOi. For the $\delta^{18}O_{pw}$ datasets the behaviour of the relationships to the wNAOi is similar to the $\delta D_{pw}$ datasets. The 3-month $\delta^{18}O_{pw}$ datasets from Grimsel (#18) and Patscherkofel (#27) show a relationship to the wNAOi but all other stations exhibit no or only weak relationships to the wNAOi (Figure 3a). The 6-month datasets from Längenfeld (21) and St. Peter (31) have a

weak relationship to the wNAOi (Figure 3b). The 6-month $\delta^{18}O_{pw}$ datasets from Villacher Alpe and Graz also show weak relationships to the wNAOi.

The $\delta^{18}O_{pw}$, $\delta D_{pw}$, temperature and precipitation datasets were also grouped according to wNAOi class as previously for the non-Alpine stations (Section 4.1). A detailed analysis of the $\delta^{18}O_{pw}$ and $\delta D_{pw}$ values of all Alpine stations shows that the median $\delta^{18}O_{pw}$ values become more negative for higher altitudes, irrespective of the wNAOi class. This observation is





well known as the "altitude effect" (Dansgaard, 1954;Schürch et al., 2003). However, there is no obvious relationship (p>0.1 for all datasets) between the class of the wNAOi and the altitude effect for the Alpine stations (Fig. 7a and 7b).

On average, the altitude effect is -0.32 ‰/100m (6-month average) and -0.30 ‰/100m (3-month average) for $\delta^{18}O_{pw}$ and -2.55 ‰/100m (6-months average) and -2.28 ‰/100m (3-months average) for $\delta D_{pw}$ (Fig. 7a and 7b). Furthermore, the
recorded air temperature at the Alpine stations change on average by about -0.58 and -0.56 K/100m for the 6-month and 3-month average respectively, independent of the wNAOi class (p>0.1 for all datasets) (Fig. 7c). The mean values of the temperature-altitude relationship correspond approximately to the moist adiabatic lapse rate. Hence, a strong relationship between $\delta^{18}O_{pw}$ and $\delta D_{pw}$ values and air temperature is observed in the Alpine stations (e.g. Schürch et al., 2003). No relationship between rainfall amount and $\delta^{18}O_{pw}$ was observed (Fig. 7d).


Because there is no relationship between the lapse rate and and/or precipitation amount with the wNAOi class, we conclude that the observed relationships between $\delta^{18}O_{pw}$ and $\delta D_{pw}$ and the class of the wNAOi for our selection of Alpine stations are (i) caused by different air mass origins linked to wNAOi states and (ii) downstream effects of the varying central European continental effect that causes more negative $\delta^{18}O_{pw}$ and $\delta D_{pw}$ values for more negative wNAOi classes. The weak or absent
relationships for stations at the Alpine divide can be caused by air masses of different origins (e.g. variable influences of the Mediterranean sourced moisture). However, detailed back trajectories of rainfall events for the entire Alpine region would be required to further evaluate this explanation. Our analysis indicates that the Alpine divide exerts an important influence on the winter hydrological cycle in the region, with precipitation north of the Alps sourced by atmospheric moisture originating from central Europe. In winter, this residual atmospheric moisture is already depleted in $^{18}O$ and $^{2}H$ when it reaches the
northern part of the Alps, reflecting the ambient winter mode of the wNAOi, thereby determining the degree of the depletion in $^{18}O$ and $^{2}H$ in north Alpine winter precipitation.

To complete the above conclusion on the mixing of atmospheric moisture for stations at the Alpine divide, the $\delta^{18}O_{pw}$ and $\delta D_{pw}$ of circum-Mediterranean stations were also analysed for their dependence on the wNAOi (a discussion of the NAO-relationships between $\delta^{18}O_{pw}$ and $\delta D_{pw}$ and temperature and precipitation can be found in the supplementary
information). These stations are Avignon (34) (southwest of the Alps), Locarno (35) (south of Grimsel), Genoa (Setri) (south of the Alps) (36) and Zagreb (37) (southeast of the Alps) (Figure 1). For the 3-month $\delta^{18}O_{pw}$ and $\delta D_{pw}$ data only those from Avignon and Zagreb shows a strong relationship to the wNAOi (about 1 ‰/wNAOi unit for $\delta^{18}O_{pw}$) (Figure 3a and 3c). For the 6-month $\delta^{18}O_{pw}$ and $\delta D_{pw}$ data, only the $\delta^{18}O_{pw}$ dataset from Locarno and the $\delta^{18}O_{pw}$ and $\delta D_{pw}$ datasets from Zagreb show a relationship to wNAOi. The NAO-relationships of $\delta^{18}O_{pw}$ and $\delta D_{pw}$ for the 3-month winter period from the Mediterranean
stations Locarno and Genoa (Setri) show that the NAO-fingerprint, which is observed for Alpine stations north of the Alpine divide, is not transferred to these two stations. The situation might change for $\delta^{18}O_{pw}$ from Locarno for the 6-month winter period where a relationship to the wNAOi is observed. The stronger relationship for this winter period could be caused by an increase of precipitation that results from air masses from Central Europe. For Zagreb, it is difficult to explain the observed



relationships between $\delta^{18}O_{pw}$ and $\delta D_{pw}$, because the closest Alpine stations show no relationship to the wNAOi. To further

investigate the mechanism that control the $\delta^{18}O_{pw}$ and $\delta D_{pw}$ datasets in Avignon and Zagreb, the origin of the air masses in dependence on the wNAOi needs to be investigated further using isotope enabled regional climate models to better constrain the effect of local temperature and precipitation on $\delta^{18}O_{pw}$ and $\delta D_{pw}$. In summary, the variable wNAO-relationship of $\delta^{18}O_{pw}$ and $\delta D_{pw}$ datasets from the investigated Mediterranean stations support the observation of that the Alpine divide represents an important boundary region of the oxygen and hydrogen isotope system of Alpine precipitation.

**5 Implications for speleothem-based and other paleoclimate reconstructions and application to a reconstruction of Holocene longitudinal speleothem $\delta^{18}O$ gradients**

The results have important implications for palaeoclimate archives that record the $\delta^{18}O_{pw}$ and $\delta D_{pw}$ values of winter precipitation (October to March, the main period of infiltration in central Europe due to low evapo-transpiration). Potentially, such archives include speleothems and ground water. The following discussion focuses on speleothem carbonate

$\delta^{18}O$ records, but is also applicable to speleothem fluid inclusion $\delta^{18}O$ and $\delta D$ records, and is relevant for other palaeoclimate archives (e.g. stable isotopes of tree rings records).

For a paleoclimate reconstruction that is based on a single speleothem from a cave site from central Europe or north of the Alpine divide, $\delta^{18}O_{pw}$ and $\delta D_{pw}$ values are typically lower for more negative wNAOi values (Figure 3). The median sensitivity of $\delta^{18}O_{pw}$ from all continental stations is 0.57 ‰/wNAOi unit. Hence, a persistent change of the average wNAOi

from e.g. +1 to -1 (i.e. -2 wNAOi units) would result in a reduction of the average $\delta^{18}O_{pw}$ by 1.14 ‰. Furthermore, monthly average air temperatures are generally in phase with the wNAOi changes, resulting in a positive linear relationship between $\delta^{18}O_{pw}$ and $\delta D_{pw}$ and air temperatures. Hence, air temperatures tend to be lower depending on the cave location, if the average wNAOi is smaller. This temperature-relationship was recently used to reconstruct historic $\delta^{18}O_{pw}$ values (Mischel et al., 2015) and its use has been suggested for wNAOi reconstructions (Casado et al., 2013). Crucially, however, the $\delta^{18}O_{pw}$

and $\delta D_{pw}$ variability is controlled not only by changes in air temperature, but also by changes in the air mass precipitation history (Section 4). Hence, the use of speleothem $\delta^{18}O$ values (or $\delta^{18}O$ and $\delta D$ values from fluid inclusions or ground water) to reconstruct the past variability of the wNAOi or winter temperatures should be undertaken cautiously, because past changes of the hydrological cycle could result in a relationship between $\delta^{18}O_{pw}$ and temperature that differs from the present day. This is particularly important for long-term reconstructions of atmospheric circulation. As for the case of individual

speleothem $\delta^{18}O$ records, caution should be exercised when interpreting changes in $\delta^{18}O$ gradients inferred from multi-speleothem regression analysis, because the continental precipitation $\delta^{18}O_{pw}$ and $\delta D_{pw}$ gradients are controlled by both the air temperature gradient and the air-mass precipitation history (Section 4).

The results from the sensitivity analysis of $\delta^{18}O_{pw}$ and temperature allow theoretically estimates of the possible influences of changes in $\delta^{18}O_{pw}$ and temperature for wNAOi changes on speleothem $\delta^{18}O$ records. For this exercise we use

the station data of Stuttgart (Canstatt) (#8), which is located in southern Germany and whose $\delta^{18}O_{pw}$ sensitivity to the





wNAOi equates approximately to the median of all continental stations. For the period from October to March, the NAO-temperature relationship for this station suggests a sensitivity of 0.63 K/wNAOi unit, and an intercept of 4.38 °C (r=0.57); no NAO-relationship is observed for the precipitation. In the example above, a decrease in the NAO index from +1 to -1 would cause a change of -1.20 ‰ in $\delta^{18}O_{pw}$ in Stuttgart precipitation, with a concomitant decrease in air temperature of 1.26 K.

Assuming that the change in $\delta^{18}O_{pw}$ and air temperature is transmitted into a cave, we can estimate both effects on speleothem $\delta^{18}O$ values. Note that our assumption about the temperature change represents a limiting case, because it implies that the annual air temperature, to which the cave air temperature is usually equilibrated, also decreases by the same value. Applying the temperature sensitivity of the equilibrium oxygen-isotope fractionation factor of Kim and O'Neil (1997) of about -0.22 ‰/K, the temperature change would result in an increase in speleothem $\delta^{18}O$ by about +0.28 ‰, but the

simultaneous decrease by 1.20 ‰ in drip water $\delta^{18}O$ would dominate, resulting in an overall decrease in the speleothem $\delta^{18}O$ by about 0.92 ‰. Therefore, the changes of precipitation $\delta^{18}O_{pw}$ would dominate the speleothem $\delta^{18}O$ record. We stress that these conditions are rarely met in natural cave systems where other processes typically influence the speleothem proxy values. Nonetheless, this exercise emphasises that the sensitivity of $\delta^{18}O_{pw}$ to wNAOi is large and that persistent changes (i.e. centennial and millennial changes) in the mean state of the wNAO are likely to produce detectable changes in

speleothem $\delta^{18}O$ in Central Europe.

It is notable that for speleothems deposited in caves close to the Alpine divide where there is little or no relationship between $\delta^{18}O_{pw}$ and the wNAOi, speleothem $\delta^{18}O$ values are likely to be dominated by the temperature-dependent isotope fractionation factor during speleothem growth rather than by changes in $\delta^{18}O_{pw}$. Hence, if modern/historic temperature

calibrations of speleothem $\delta^{18}O$ values are conducted, a relatively straightforward, but site-specific negative correlation between speleothem $\delta^{18}O$ and temperature is to be expected for sites close to the Alpine divide. This inference is supported by the two currently available studies; one at Spannagel Cave, Austria, which is located close to the Alpine divide in the central Alps where the temperature calibration is -0.44 ‰/K (Mangini et al., 2005) and one from Milandre Cave, situated in the Swiss Jura Mountains (western Alps), where the temperature calibration is 0.70 ‰/K (Fleitmann, 2016 personal

communication). Taking the temperature sensitivity of $\delta^{18}O_{pw}$ of about 0.95 ‰/K for the GNIP station Konstanz, located at the similar latitude as Milandre Cave, and the temperature sensitivity of the equlibrium isotope fractionation between water and calcite of about -0.22‰/K, this yields a nett value of 0.73‰/K. This value is consistent with the temperature calibration found for Milandre Cave.

As an example of the expected change in speleothem $\delta^{18}O$ that would result from a persistent wNAOi change with associated

changes in $\delta^{18}O_{pw}$ and temperature, the calculated slopes for the 6-month (October-March) longitudinal $\delta^{18}O_{pw}$ gradients can be converted into expected speleothem $\delta^{18}O$ gradients. Such an estimate requires that both the longitudinally variable winter temperature gradients (Figure 5a), as well as the $\delta^{18}O_{pw}$ slopes (Figure 4a) are taken into account. The effects of varying temperature gradients are considered via the temperature-dependent oxygen isotope fractionation between water and calcite. This estimate shows that the slope of the longitudinal speleothem $\delta^{18}O$ gradient is largest for the highest wNAOi class and





decreases for lower wNAOi classes. The comparison between the expected slopes and a recent reconstruction of speleothem based $\delta^{18}O$ gradients throughout the Holocene (McDermott et al., 2011) shows remarkable agreement with these calculated values (Figure 8).

Assuming that the present-day mechanisms that determine the observed relationships between the wNAOi and temperature, precipitation and $\delta^{18}O_{pw}$ are relevant for the boreal winter of the entire Holocene, the evolution of the

reconstructed Holocene speleothem $\delta^{18}O$ gradients can be interpreted as reflecting predominantly negative wNAOi values in the Early Holocene and mainly positive wNAOi values in the late Holocene winters. For this assumption (NAO type forcing) the changes of the speleothem $\delta^{18}O$ gradients are caused by changing temperature gradients (steeper west to east European temperature gradients in the early Holocene) and a changing precipitation history (increased precipitation and/or reduced amount of moisture in the atmosphere).

**6 Conclusions and Summary**

This study investigated the relationships between central European and Alpine $\delta^{18}O_{pw}$ and $\delta D_{pw}$ values and the North Atlantic Oscillation index for the boreal winter (wNAOi). $\delta^{18}O_{pw}$ and $\delta D_{pw}$ data for 37 meteorological stations distributed over central Europe and the Alps were analysed. This study demonstrates that the European continental isotope effect depends on the wNAOi, with steeper gradients associated with more negative wNAOi values. Hence, precipitation $\delta^{18}O_{pw}$

and $\delta D_{pw}$ values across central Europe are more negative for lower wNAOi winters. We argue that the strength of this change is caused by (i) a steeper west to east continental air temperature gradient and (ii) a decrease in the precipitable water content of the atmosphere during more negative wNAOi conditions. An evaluation of the longitudinal $\delta^{18}O_{pw}$ and $\delta D_{pw}$ gradients from an isotope-enabled general circulation model, ECHAM5-wiso, shows that the simulated slopes are well reproduced compared with those reconstructed from observational data. The mechanisms that cause the variability of the

simulated slopes in the ECHAM5-wiso output are steeper compared with the observational datasets and suggest that the ECHAM5-wiso simulated $\delta^{18}O$ and $\delta D$ values are largely controlled by the air temperature gradient, but this requires further investigation.

The emerging picture for Alpine stations is that for stations north of the alpine divide, $\delta^{18}O_{pw}$ and $\delta D_{pw}$ values show a similar relationship to the wNAOi, as continental stations north of the Alps. However, in contrast with the processes that

determine $\delta^{18}O_{pw}$ and $\delta D_{pw}$ values at continental stations, $\delta^{18}O_{pw}$ and $\delta D_{pw}$ characteristics at Alpine stations are mainly a downstream residual effect of the changing continental effect. Furthermore, $\delta^{18}O_{pw}$ and $\delta D_{pw}$ values of stations close to the Alpine divide in the central and eastern Alps exhibit a weak or absent relationship to the wNAOi. The results of this study have important implications for palaeoclimate reconstructions that are based on winter $\delta^{18}O_{pw}$ and $\delta D_{pw}$ values, such as speleothem $\delta^{18}O$ records.



**Data availability**

The processed data is available through supplementary material. Unprocessed data can be accessed through http://www-naweb.iaea.org/napc/ih/IHS_resources_gnip.html and https://wasser.umweltbundesamt.at/h2odb/.

**Acknowledgement**

We thank all operating organizations of the Global Network of Isotopes in Precipitation (GNIP) and the Austrian Network of
Isotopes in Precipitation (ANIP) and all contributing researchers for their efforts to run and maintain the stations, which
made this study possible. MD is funded by the Irish Research Council (IRC) by a Government of Ireland Postdoctoral
Fellowship (GOIPD/2015/789). FMcD acknowledges support from Science Foundation Ireland through its Research
Frontiers Program (RFP) Grants 07/RFP/GEOF265 and 08/FRP/GEO1184.

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





**Tables**

| | W-E difference for $\delta^{18}O_{pw}$ | | | | | | | | W-E difference for $\delta D_{pw}$ | | | | | | | |
|---|---|---|---|---|---|---|---|---|---|---|---|---|---|---|---|---|
| | Observed | | Calculated | | Deviation | | | | Observed | | Calculated | | Deviation | | | |
| wNAOi class | 6m | 3m | 6m | 3m | **6m** | | **3m** | | 6m | 3m | 6m | 3m | **6m** | | **3m** | |
| | ‰ | ‰ | ‰ | ‰ | **‰** | *%* | **‰** | *%* | ‰ | ‰ | ‰ | ‰ | **‰** | *%* | **‰** | *%* |
| I: 1.77 | 4.78 | 6.02 | 3.75 | 4.98 | **1.03** | *22.5* | **1.04** | *17.3* | 42.62 | 52.10 | 32.80 | 43.56 | **9.82** | *23.0* | **8.54** | *16.4* |
| II: 1.18 | 4.94 | 5.14 | 3.28 | 3.98 | **1.66** | *33.6* | **1.16** | *22.6* | 44.06 | 46.19 | 28.72 | 34.84 | **15.34** | *34.8* | **11.35** | *25.8* |
| III: 0.40 | 5.73 | 6.32 | 4.00 | 5.54 | **1.73** | *30.2* | **0.79** | *12.5* | 49.37 | 51.68 | 35.01 | 48.44 | **14.36** | *29.1* | **3.24** | *6.3* |
| IV: -0.46 | 6.66 | 7.89 | 4.18 | 5.58 | **2.48** | *37.2* | **2.31** | *29.3* | 56.93 | 65.10 | 36.58 | 48.81 | **20.35** | *35.7* | **16.29** | *25.0* |
| V: -1.17 | 6.22 | 7.38 | 4.44 | 6.55 | **1.79** | *28.8* | **0.83** | *11.2* | 53.44 | 61.11 | 38.82 | 57.31 | **14.62** | *27.4* | **3.80** | *6.2* |
| VI: -2.23 | 6.54 | 7.56 | 3.37 | 5.05 | **3.17** | *48.5* | **2.51** | *33.2* | 54.61 | 64.56 | 29.46 | 44.21 | **25.14** | *46.0* | **20.35** | *31.5* |

**Table 1.** Differences in $\delta^{18}O_{pw}$ and $\delta D_{pw}$ (6 month (6m) and 3 month (3m) winter periods) between the western- (Valentia, Ireland) and eastern-most station (Krakow, Poland) for all wNAOi classes based on continental GNIP station datasets. The median wNAOi for classes I to VI respectively are: 1.77, 1.18, 0.40, -0.46, -1.17 and -2.23. The empirical estimates for the
*observed* W-E difference of $\delta^{18}O_{pw}$ and $\delta D_{pw}$ presented here are based on linear regressions of the observed trends (Figure 4). The *calculated* $\delta^{18}O_{pw}$ and $\delta D_{pw}$ values are based on the temperature difference between Valentia (Ireland) and Krakow (Poland) estimated from the linear regression of the continental GNIP station temperature datasets and temperature sensitivities for $\delta^{18}O_P$ and $\delta D_P$ from Dansgaard (1964). Deviation (bold numbers) means the difference between observed and calculated $\delta^{18}O_{pw}$ and $\delta D_{pw}$ values. The italic numbers states the deviation relative to the observed difference given in %.


**Figures**





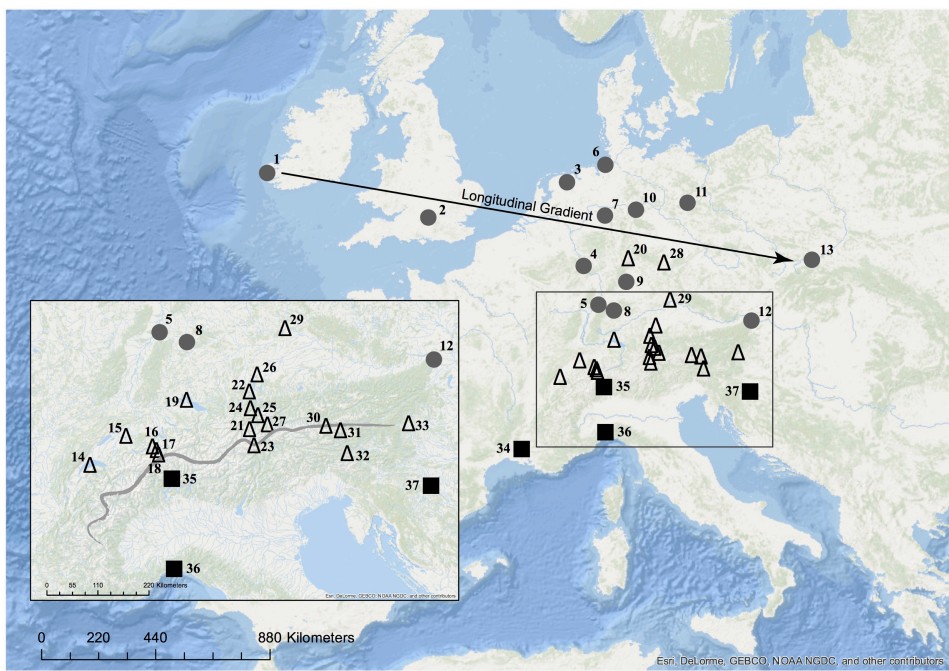

**Figure 1.** Map showing the location of all investigated stations: closed grey circles show the location of continental stations (altitude ≤350m with no Mediterranean influence); open triangles indicate alpine and high altitude (>350m) stations; closed squares show the location of 'Mediterranean influenced' stations. The grey line indicates the Alpine Divide. Station codes for continental stations (from west to east): 1) Valentia (Observatory); 2) Wallingford; 3) Groningen; 4) Koblenz; 5) Karsruhe; 6) Cuxhaven; 7) Bad Salzulfen; 8) Stuttgart (Cannstatt); 9) Würzburg; 10) Braunschweig; 11) Berlin; 12) Vienna (Hohe Warte); 13) Krakow (Wola Justowska). For high altitude stations: 14) Thonon-Les-Bains; 15) Bern; 16) Meiringen; 17) Guttannen; 18) Grimsel; 19) Konstanz; 20) Wasserkuppe-Rhoen; 21) Längenfeld; 22) Hohenpeisenberg; 23) Obergurgl; 24) Garmisch-Partenkirchen; 25) Scharnitz; 26) Neuherberg; 27) Patscherkofel; 28) Hof-Hohensaas; 29) Regensburg; 30) Böckstein; 31) St. Peter; 32) Villacher Alpe; 33) Graz Universität. For Mediterranean influenced stations: 34) Avignon; 35) Locarno; 36) Genoa (Sestri); 37) Zagreb.





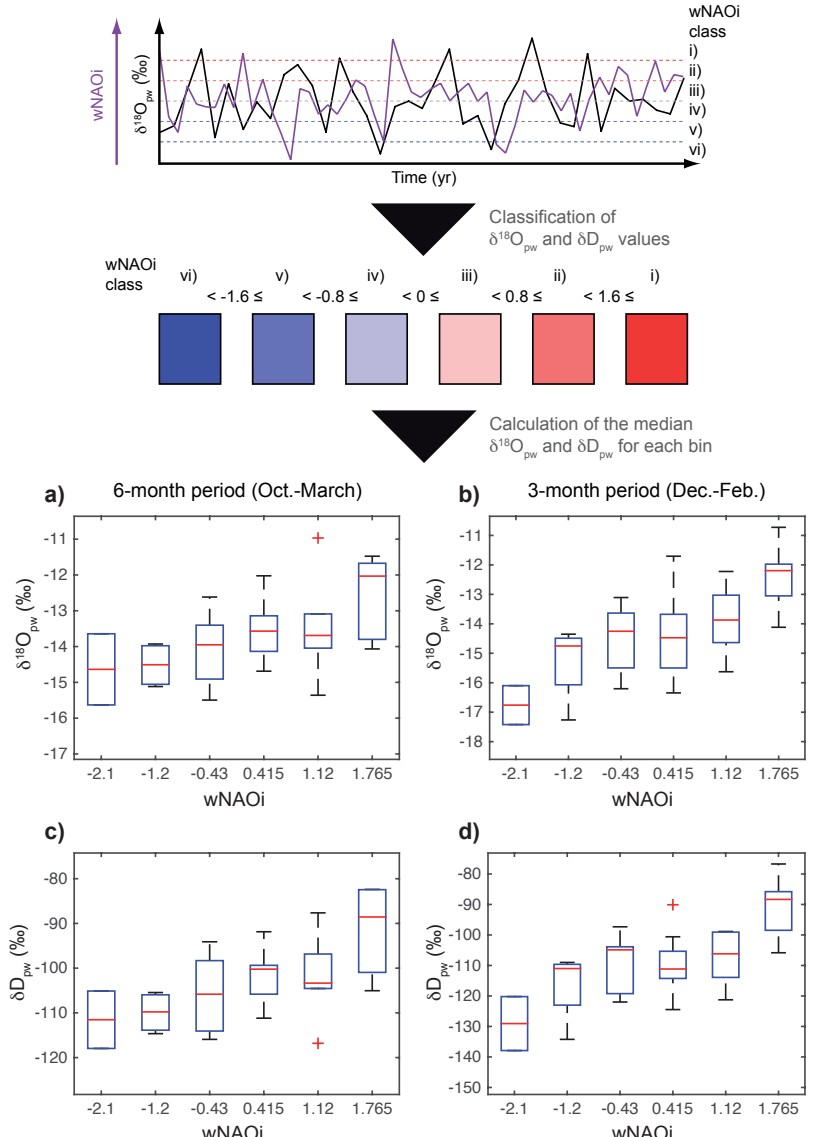



**Figure 2.** The upper part of the figure depicts how the precipitation weighted $\delta^{18}O_{pw}$ and $\delta D_{pw}$ values are classified here into

six classes depending on winter NAO index. Median values are calculated for every class, and this is used for further analysis. Panel a) to d) show the processed results for the six classes for one exemplar GNIP station (Garmisch-Partenkirchen, Germany), for which $\delta^{18}O_{pw}$ and $\delta D_{pw}$ values have a typical sensitivity to wNAOi (see Figure 3). The left panels show the box plots for the 6 months period (October to March) and the right panels for the 3 months period (December to February). The upper panels illustrate the $\delta^{18}O_{pw}$ values, the lower panels the $\delta D_{pw}$ values. Every box plot

illustrates the statistical variables (median, min, max, 25 and 75% quantile) for every wNAOi class from lowest to highest (left to right). For the individual wNAOi classes, the red line illustrates the median of the data compilation; the edges of the blue rectangles mark the 25% and 75% quantile; the black bars illustrates the minimum and maximum values and the red cross denotes 'outliers'.






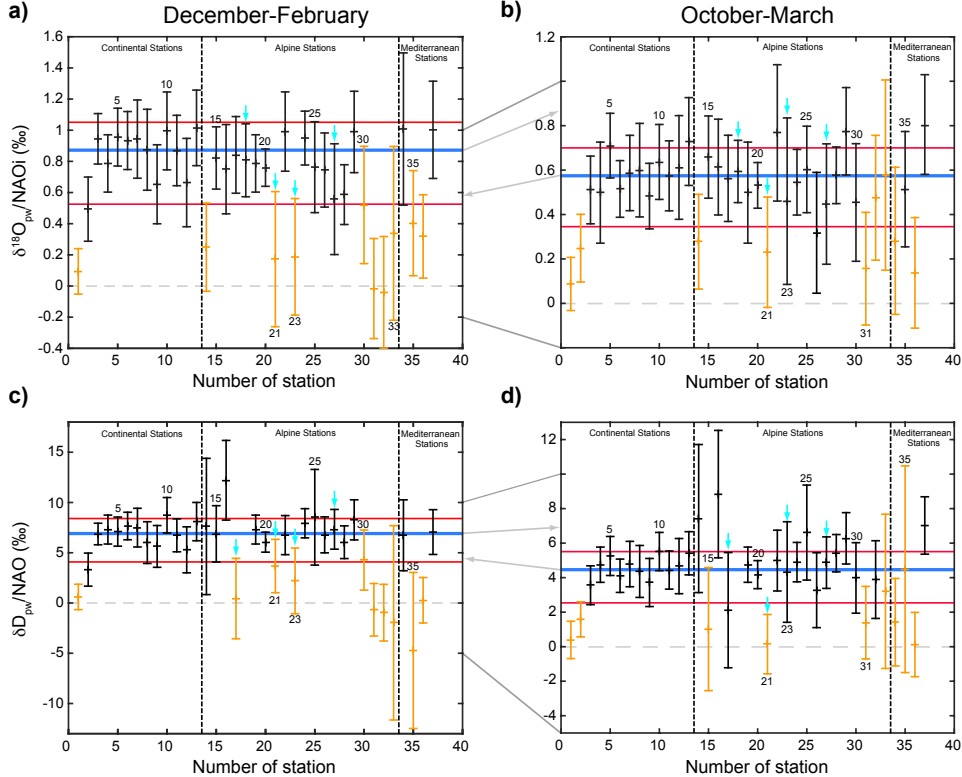

**Figure 3.** The panels illustrate the slopes of the linear regressions between yearly $\delta^{18}O_{pw}$ (a and b) and $\delta D_{pw}$ (c and d) and the wNAOi for each individual station for the 3-month winter period (December-February) (left panels) and the 6-month winter period (October-March) (right panels). If the slope is illustrated in black the linear correlation coefficient is greater than 0.3; otherwise it is shown in orange. The blue lines indicate the median values of all continental stations; the upper and lower red lines highlight the 1-sigma standard deviation around the mean value. The grey dashed lines indicate a slope of 0 (i.e. no sensitivity). Station Garmisch-Partenkirchen (24) whose sensitivity on the wNAOi classes is shown in detail in Figure 2 is typical of the investigated stations. The cyan coloured arrows indicate stations closest to the Alpine Divide. Numbers indicate station codes as in Figure 1.



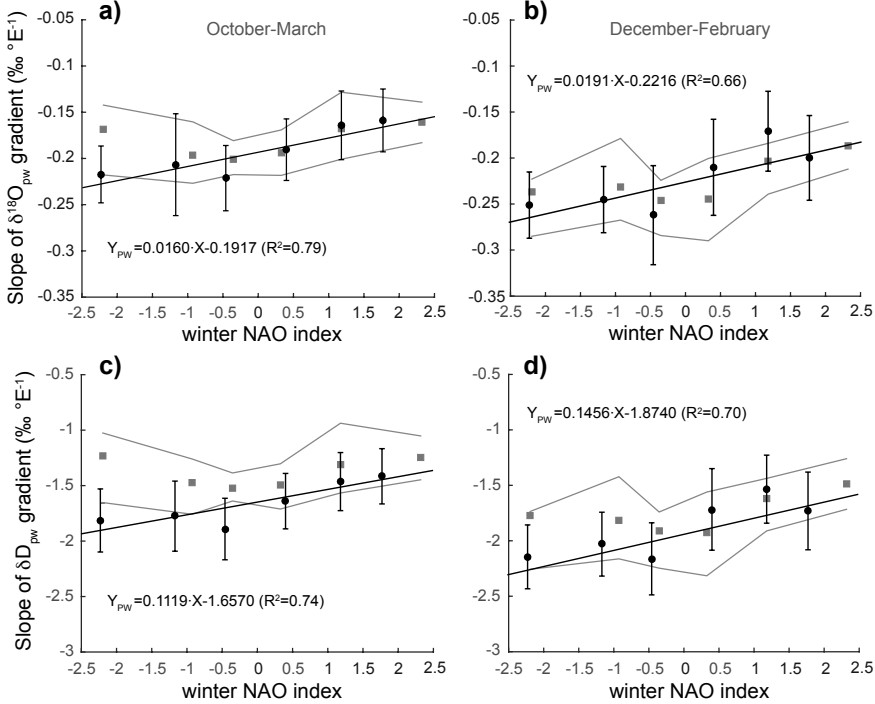

**Figure 4:** Illustration of the slopes of the $\delta^{18}O_{pw}$ (a and b) and $\delta D_{pw}$ (c and d) longitudinal gradients across Europe (filled circles) – and their respective standard errors – calculated from 13 continental GNIP stations for the 6-month (October-March) (a and c) and 3-month (December to February) (b and d) winter period. More negative wNAOi classes result in a steeper isotope gradient across Europe in winter and, therefore, more strongly depleted $\delta^{18}O_{pw}$ and $\delta D_{pw}$ values with increasing distance (towards the east) from the European western margin. The coefficient for the linear regression between the observed slopes and the class of the wNAOi is $0.016\pm0.004$ ($r^2=0.79$; $p<0.05$) for Fig. 1a, $0.019\pm0.006$ ($r^2=0.65$; $p=0.0502$) for Fig. 1b; $0.111\pm0.033$ ($r^2=0.74$, $p<0.05$) for Fig. 1c and $0.146\pm0.048$ ($r^2=0.70$, $p<0.05$) for Fig 1d (units are ‰ °E$^{-1}$/wNAOi). These equations state the results from the linear regression where $Y_{PW}$ is the slope of the $\delta^{18}O_{pw}$ or $\delta D_{pw}$ gradient and X is the wNAOi. The filled grey squares show the median slopes of the ECHAM5-wiso simulations; the grey envelope indicates the 25% and 75% quantiles of the ECHAM5-wiso slopes.





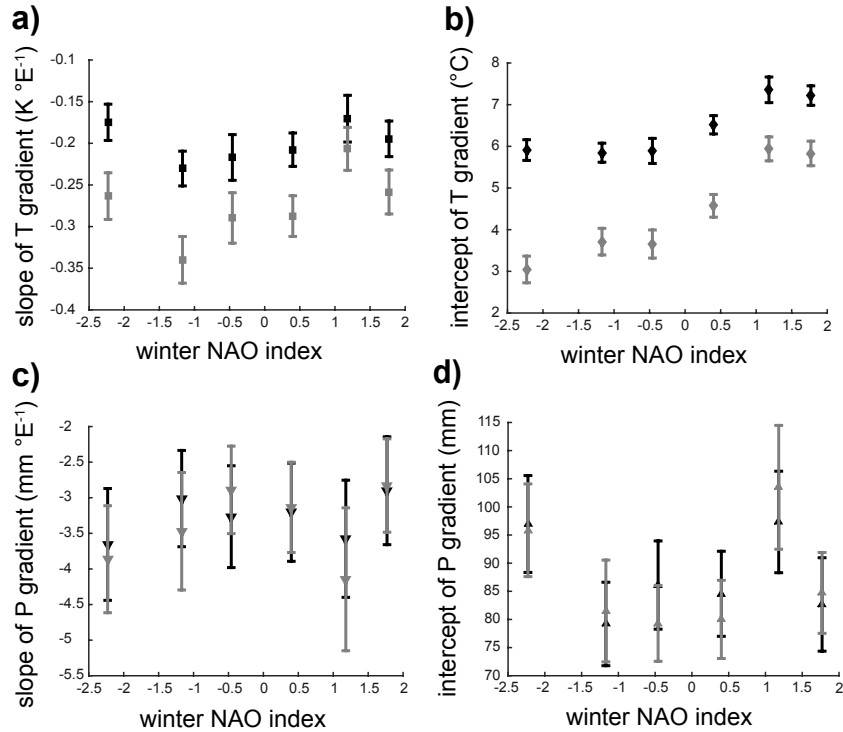

**Figure 5.** Based on observational data only, these four panels illustrate the slope of the continental gradient for (a) temperature and (c) precipitation as a function of the class of wNAOi that are calculated from GNIP station datasets. Panel b) and d) shows the intercepts of the linear regression for the continental temperature and precipitation gradients versus the wNAOi, respectively. Black symbols indicate the results for the 6-month winter period (October-March); grey symbols denote results for the 3-month (December-February) winter period. The slopes for temperature and precipitation show no relationship to the wNAOi if all six classes are analysed (p>0.1). However, omitting the most negative wNAOi class yields a significant linear correlation of 0.71 and 0.67 (p<0.01) for the 6 and 3 month averages, respectively. While there is no significant relationship between the intercept of the precipitation gradients with the wNAOi (p>0.1), the intercept of the air temperature gradients shows a clear trend, with lower temperatures associated with lower wNAOi values.





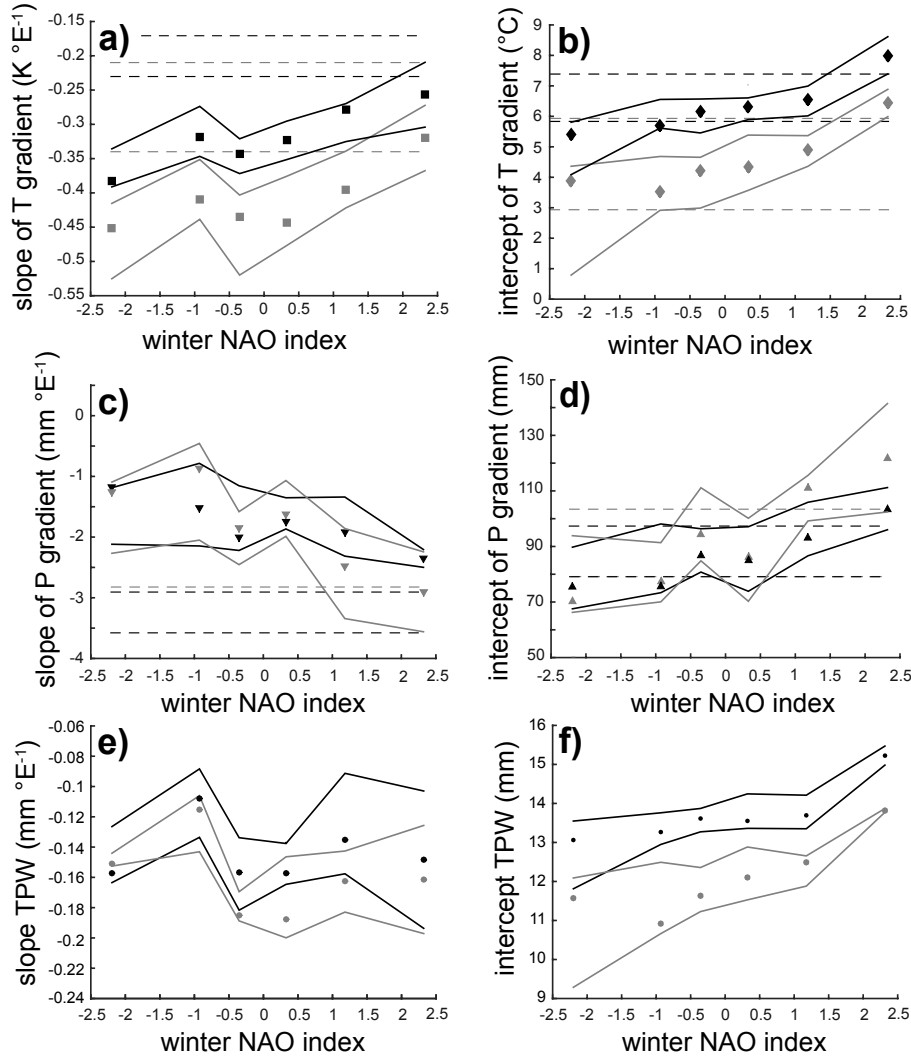

**Figure 6:** ECHAM5-wiso outputs: The six panels illustrate the median slope of the gradient for a) temperature (T), c) precipitation (P) and e) the total precipitable water (TPW) depending on the class of the wNAOi from the analysed



ECHAM5-wiso grid cells. Panel b), d) and f) shows the median intercept of the linear regression for the temperature, precipitation and total precipitable water gradient versus the class of wNAOi. The envelopes (straight line) indicate the 25% and 75% quantile. Back colours indicate the results for the 6-month winter period and grey colours the results from 3-month

660  winter period. The dashed lines in panel a) to d) indicate the observed variability of these parameters derived from the observational datasets from the GNIP stations as illustrated in Figure 5.

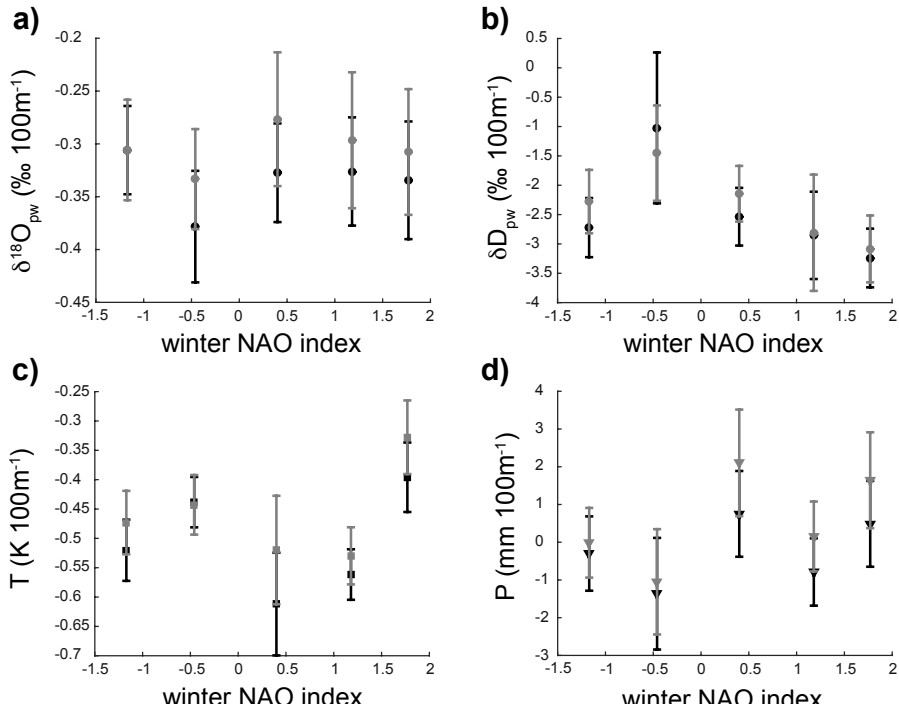

665  **Figure 7:** The four panels show the rate of change of (a) $\delta^{18}O_{pw,}$ (b) $\delta D_{pw,}$ (c) temperature (T) and (d) precipitation (P) as a function of altitude for different wNAOi classes. (The lowest wNAOi class was not analysed because data of only four stations (out of 17) is available.) All Alpine stations are included. None of the correlations are statistically significant (p>0.1). Black symbols indicate the results of the 6-month winter period; grey symbols denote the 3-month winter period.

670



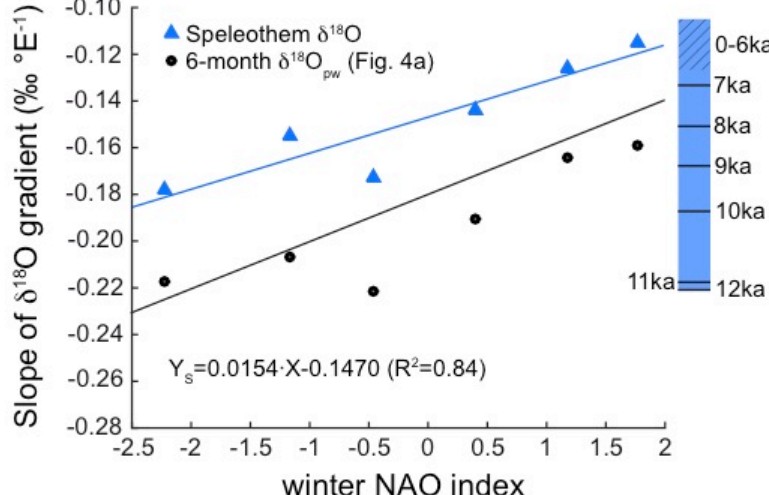

**Figure 8:** The blue triangles illustrate the forward-modelled longitudinal slopes for speleothem records calculated for each wNAOi class from their respective 6-months (October-March) longitudinal $\delta^{18}O_{pw}$ gradients (closed circles) and the observed temperature gradient converted into a speleothem $\delta^{18}O$ gradient using a sensitivity of 0.225 ‰/K (Kim and O'Neil, 1997). The blue and black line indicates the regression line from the longitudinal speleothem $\delta^{18}O$ and $\delta^{18}O_{pw}$ gradient, respectively. The blue bar highlights the range of reconstructed longitudinal speleothem $\delta^{18}O$ slopes during the Holocene (McDermott et al., 2011). The temporal evolution of the speleothem $\delta^{18}O$ slopes (McDermott et al. (2011) vary between -0.2208 ‰/°E and -0.1336 ‰/°E from 12 ka to 7 ka, and vary and between -0.1266 ‰/°E and -0.1046 ‰/°E from 6 ka to the present (shaded area). Comparison of the speleothem slopes with those calculated in this study suggests predominantly wNAO- modes in the Early Holocene and wNAO+ modes in the Late Holocene.