# Peer review of "North Atlantic Oscillation controls on oxygen and hydrogen isotope gradients in winter precipitation across Europe; implications for palaeoclimate studies"

_Climate of the Past, 2016_

## Referee Comment (RC1) · Anonymous Referee #1 · 3 Aug 2016

Summary

The authors have shown convincingly that for continental stations north of the alpine divide in western and central Europe that there is a robust relationship between the winter NAO index and the stable isotope values of precipitation. The study represents a solid advance and the use of the statistics is convincing. The paper is data-dense and well written. Further, the observations of the continental gradients over time fits with the previously-published idea of looking at speleothem-based d18O profiles of stalagmites or other proxies to reconstruct NAO in the past. The observations are also

somewhat substantiated by isotope-enabled model results.

General Comments The NAO is a powerful mechanism to explain different Holocene isotope gradients. In this sense the main conclusion of the paper is sound, at least for essentially 'modern' conditions for which there are precipitation isotope measurements. Under late Holocene boundary conditions of restricted ice extent, near-modern insolation, and near-modern vegetation cover, the NAO-isotope relationship is likely valid. But for how long into the past? I would feel comfortable extending it a few millennia, perhaps five. But the early Holocene represents a significantly different climate than modern because of the ice sheets and a different orbital configuration. I would suggest that the assumption of time-stationarity back to the early Holocene of the wNAOi and d18Opw is possibly correct but currently unsupported. The two sentences around Line 460 do not do this point justice. The manuscript would be stronger by either supporting the assertion with additional evidence that, say, the NAO existed more or less as we know it know during the early Holocene NAO, when the presence of upstream ice sheets and different insolation and vegetation regimes were present, or by providing stronger caveats for extending the NAO discussion prior to the late Holocene. Perhaps using the term "NAO-like" instead of the NAO, while emphasizing that early-Holocene climate had distinctly different forcings and boundary conditions than the late Holocene, would be advisable.

One other conclusion is pretty easily testable but doesn't seem to have been evaluated rigorously: that precipitable water is less during negative wNAOi states. The current study would have much stronger standing with readers if estimates of precipitable water (say from the NCEP database) and wNAOi were compared directly.

Specific Comments

Line 122: change to "more strongly negative".

Please give the altitudes of the >350m non-alpine stations. If they don't differ much from the <350m stations (line 134), then why separate them out? Would it be better to

include them with the <350 m stations because of similar response to the NAO?

Line 296: replace "confirmed" with "supported". From a semantics point of view, observations can "confirm," but models, being not real, can only support.

Line 337: change exceptions to exception, or provide another example.

Line 371-372: back trajectory analysis would be really interesting to do, also for the stations north of the divide, as it may explain some of the variable strengths of the d18O/longitude relationship for the NAO classes. I think this point deserves a little more emphasis either in the results section, or later in the discussion section. I could see it being relevant for high- vs. medium-latitude North Atlantic oceanic source regions for different moisture sources advected inland.

Line 418: Theoretical not theoretically

Lines 411-417: the point about modern relationships maybe not being representative of past conditions is important and requires some more emphasis.

Line 443: would read better as "situated in the Swiss Jura mountains approximately xxx km from the alpine divide..."

Line 446: equilibrium typo; line 446 "net" not "nett"

Line 457: There is no Figure 8 in the manuscript. Supplemental Figure? I would like to such a figure in the main text, as it is a crucial test of the current manuscript's hypothesis.

Line 458-460: this is where I would suggest the assumption of stationarity of the wNAOi and d18Opw is not supported. Certainly not for the "entire Holocene", but probably true for the past few millennia or so after ice sheets had decayed and land vegetation was established. One way around this problem is, for the pre-late Holocene, to refer to "NAO-like" behavior.

---

## Referee Comment (RC2) · Anonymous Referee #2 · 4 Aug 2016

This manuscript presents an interesting perspective on the modern NAO, and then uses this information to reconstruct its variability through the Holocene. The conclusions of the manuscript are important, and the manuscript is well written with only a few grammatical/spelling errors. The manuscript could almost be accepted as is, but I have a few general comments that I would suggest that the authors consider. It may be that I have missed some obvious points, in which case a slight clarification or simply a response to the comment would suffice. However, I think that a thorough consideration of at least some of the points will help improve the impact of the final published paper.

[Figure]

Comment #1: It is not clear to me that the ECHAM5-wiso simulations add anything substantial to the manuscript. They seem to me to interfere with the flow and the communication of the main points of the research. The research does not provide a thorough test of the model, or a comparison with other existing models (both are beyond the scope of the manuscript). Because the manuscript uses measured data (e.g., NAO data, GNIP data), there exists little uncertainty regarding the quality of the data or the synoptic conditions associated with the data. So why include the model results, which are considerably less certain than the actual data, and in fact are sometimes inconsistent with the other results, and presumably wrong? I would suggest either i) remove all mentions to the model or ii) move the model and associated discussion to the supplemental material. The strongest reason for keeping the model in is the independent analysis of the amount of precipitable water in the atmosphere - so maybe Figure S2 could be moved to the main text, with the 13 main GNIP sites marked on it, and the rest of the model discussion moved to the supplemental material. I think that a reference to the Supplemental Material would suffice as an explanation of how the new figure (the old S2) was constructed.

Comment #2: I wonder if a spatially stationary gradient can satisfactorily capture the complexity of the NAO. In particular, I suggest that the authors consider the 'Augmented NAO Index' of Wang et al., 2012, GRL within the context of their own work. The presumed stationarity of the pressure systems defining the NAO is an issue with most NAO studies, but because this study focuses on a spatial gradient perhaps it should focus more attention on this issue. Would the longitudinal gradient not change orientation through time as pressure centres shift location? Particularly in the Early Holocene, where more extensive sea ice would have relocated the pressure centers?

Comment #3: How dependent are the gradients calculated in Figure 4 on Valentia and Wallingford? Would the gradients be uniformly flat if these two sites were omitted? Even if this is the case, it does not imply a problem with the conclusions, though I suggest the authors investigate whether or not the gradients are driven by these two

sites. Also, should these be referred to as 'continental sites' or as 'maritime sites' particularly since they do seem to behave very differently than the other sites?

Minor comments: Line 23: 'analyzed extensively' instead of 'with great effort', which suggests the analyses themselves were difficult. 82: how was it determined that these stations had no Mediterranean influence? 447: 'nett' 468: 'Central Europe' or 'central Europe' – be consistent

---

## Referee Comment (RC3) · Anonymous Referee #3 · 6 Aug 2016

General comment:

Deininger et al. present an interesting manuscript demonstrating the influence of the NAO on the d18O and dD values of winter precipitation in Europe as well as the West-East gradient across Europe, i.e., the strength of the continental effect in dependence of the NAO. Comparison with the data of an isotope enabled climate model (ECHAM5-wiso) show a broad agreement with the meteorological data and enables to study the underlying processes in more detail. The evaluation suggests that the dependence of the d18O and dD values on the NAO results from both variable air temperature and

amount of precipitable water in the atmosphere. These are important findings, which should definitely be published. Finally, the authors discuss the potential to reconstruct past NAO variability (or at least persistent states of the NAO) based on changes in the gradients across Europe.

The paper is well written, and the complicated calculations (dependence of the slope on NAOi) are presented in a way that even non-expert readers should be able to understand the main messages. The results appear robust and most of the conclusions seem justified to me. Thus, I can definitely recommend publication in Climate of the Past.

I have a few editorial and specific comments (listed below), which should be taken into account prior to publication.

Detailed comments:

Line 40 ff.: The second paragraph of the introduction is relatively long and mainly summarises what will be shown and discussed in the paper. This section could be shortened substantially to make the paper more concise.

Line 46: "...to better evaluate the NAO-dependence on isotope longitudinal gradients." I thought the gradient depends on the state of the NAO and not the other way round? This should be clarified here and throughout the paper.

Line 56: 37 stations have been analysed. 28 are GNIP stations, 6 are ANIP stations. What about the remaining 3?

Line 126 ff.: "Comparison of the longitudinal $\delta18O_{pw}$ and $\delta D_{pw}$ gradients derived from the ECHAM5-wiso with those from the station-based data show that slopes from the ECHAM5-wiso data reproduce the observed station-based slopes quite well (Figure 4)." I do not agree with this statement. All model data sets show a curve (i.e., the most negative slopes are shown for the 3rd and 4th NAO class) rather than a linear relationship with the NAO classes. It is, thus, misleading to state that the model data

reproduce the station data "quite well". It would be good to see the fit statistics (slope, r2 and p-value) not only for the station data, but for the model data as well (compare caption of Fig. 4).

Line 212 ff.: "Repeating the calculations using the vapour-ice phase change (snow) instead results in calculated differences that are still too small to explain the observed differences in $\delta$18Opw and $\delta$Dpw between the western- and eastern-most stations (not shown)." Please provide a bit more information on this. I do not request a detailed discussion, but in the present form, the reader would not be able to do the calculations themselves if they wanted to.

Line 217 ff.: "By contrast with the observational (GNIP) data discussed above, the ECHAM5-wiso simulated differences in $\delta$18Opw and $\delta$Dpw can largely be accounted for by model air-temperature differences alone." ... "Overall, however, these results supports the conclusion above that the winter air temperature effect on the longitudinal winter $\delta$18Opw and $\delta$Dpw gradients is insufficient to explain the observed difference between the western and eastern GNIP stations." This is not clear to me and even appears contradictory. If the effects observed in the model data can be explained by the model temperatures, this does not support the conclusion derived from the data. Please clarify.

In addition: "It is intriguing that the observed (GNIP datasets) and simulated (ECHAM5-wiso simulations) temperature slopes differ (Figure 6), while the slopes for longitudinal $\delta$18Opw and $\delta$Dpw gradients are apparently similar (Figure 4)." As far as I understood the text, the model temperatures in Eastern Europe are colder than those of the station data. If the d18O and dD values (i.e., the gradient and its dependence on the state of the NAO) are similar in the model and the data, but the temperatures are different, this either means that the dependence of model temperature on the NAO is too strong or that the sensitivity of the model d18O and dD values is too weak. In any case, this is an important difference, which makes it difficult to use the model data to interpret the station data. Based on the discussion following below, however, this statement is not

[Figure]

necessary.

Line 330 ff.: "The reason for the different strength of these two mechanisms (temperature gradient and precipitation history) on the longitudinal $\delta 18Opw$ and $\delta Dpw$ gradients for the observed (GNIP) and simulated (ECHAM5-wiso) datasets remains unclear, suggesting that the ECHAM5-wiso simulations warrant further investigation." I am not a climate modeller, but how representative are the precipitation data of the model for the 13 stations (still a relatively low number) considered here. As far as I know, simulating (high-resolution) precipitation patterns is still difficult. Thus, the model data (which represent climate variability in a larger grid cell) may be more representative for the dependence of the west-east gradient in precipitation on the NAO than the station data. In summary, it is not very surprising for me that the precipitation data of the stations do not show a dependence on the state of the NAO, but the model data do.

Line 334 ff.: I would suggest to strongly shorten the first paragraph of section 4.2. It only summarises results from the analysis, which has partly already been presented in section 3.2. I would move all these results in section 3.2, and briefly summarise the findings here in one or two sentences.

Line 401 ff.: I would remove the reference to tree rings here, which mainly record summer climate (and water isotopes).

Line 426 ff.: "Note that our assumption about the temperature change represents a limiting case, because it implies that the annual air temperature, to which the cave air temperature is usually equilibrated, also decreases by the same value." As the authors state themselves, this assumption is not reasonable. It may be possible to find a correlation between winter and annual temperature (in the station and the model data). Based on that, one could try to estimate the dependence of annual temperature on the state of winter NAO. However, temperature is very stable in most caves, and an inter-annual change of 1.3 °C, as assumed in the example for the station Stuttgart, is almost impossible and may only occur in a strongly ventilated cave. In such caves, however,

other effects, such as precipitation of CaCO3 under conditions of disequilibrium stable isotope fractionation or evaporation, will probably dominate the d18O values of the speleothem. The temperature effect may only be visible on a decadal or even longer time scale. Thus, the reference to persistent changes in the NAO on centennial to millennial time scales should be given at the beginning of the paragraph. However, I would rather suggest to remove the calculation because the caveats may not be present to many readers.

Section 5 in general: Since this section discusses the potential of speleothems for an NAO reconstruction based on speleothems, I miss a critical discussion of other potential "problems" of speleothems for NAO reconstruction (smoothing of the signals in the aquifer, contributions of different seasons than winter, disequilibrium stable isotope fractionation, dating uncertainties, etc.). I know that the authors are aware of these problems, so they should not be omitted from the discussion here. Mischel et al. (2015) have modelled some of these processes in detail. Their study could be referenced in this context.

Finally, as the two other reviewers question the stationarity of the NAO and the gradient in the past and, in particular, its dependence on the location of the pressure systems during the Early Holocene, it may be interesting to read and discuss the recent paper by Wassenburg et al. (2016).

References

Mischel, S. A., Scholz, D., and Spötl, C., 2015. d18O values of cave drip water - a promising proxy for the reconstruction of the North Atlantic Oscillation? Climate Dynamics 45, 3035-3050.

Wassenburg, J. A., Dietrich, S., Fietzke, J., Fohlmeister, J., Jochum, K. P., Scholz, D., Richter, D. K., Sabaoui, A., Spötl, C., Lohmann, G., Andreae, M. O., and Immenhauser, A., 2016. Reorganization of the North Atlantic Oscillation during Early Holocene deglaciation. Nature Geoscience 9, 602-605.

---

## Author Comment (AC1) · 1 Oct 2016

General Comments: I would suggest that the assumption of time-stationarity back to the early Holocene of the wNAOi and d18Opw is possibly correct but currently unsupported. The two sentences around Line 460 do not do this point justice. The manuscript would be stronger by either supporting the assertion with additional evidence that, say, the NAO existed more or less as we know it know during the early Holocene NAO, when the presence of upstream ice sheets and different insolation and vegetation regimes were present, or by providing stronger caveats for extending the NAO discussion prior

to the late Holocene. Perhaps using the term "NAO-like" instead of the NAO, while emphasizing that early-Holocene climate had distinctly different forcings and boundary conditions than the late Holocene, would be advisable.

We will revise this section and extend the discussion to justify the points raised by the reviewer. For this we will include recent studies that investigate the stationarity of the NAO during the Holocene (Wassenburg et al., 2016;Walczak et al., 2015). Furthermore, we will extend the discussion on the stationarity of the NAO during the Holocene in general and highlight potential caveats of our approach.

One other conclusion is pretty easily testable but doesn't seem to have been evaluated rigorously: that precipitable water is less during negative wNAOi states. The current study would have much stronger standing with readers if estimates of precipitable water (say from the NCEP database) and wNAOi were compared directly.

We will revise the manuscript as suggested. We will include the analysis of NCEP/NCER reanalysis data of precipitable water in the manuscript. We will evaluate the dependence of the precipitable water on the wNAOi over Europe and will add a new figure in the manuscript. The NCEP/NCER reanalysis data shows a similar variability as shown for the ECHAM5-wiso data. Therefore, the results and conclusions drawn from the ECHAM5-wiso data are also valid for the NCEP/NCER reanalysis and the overall conclusions don't change. (See the attached figure 1.)

Specific Comments: Line 122: change to "more strongly negative". We will revise the manuscript as suggested.

Please give the altitudes of the >350m non-alpine stations. If they don't differ much from the <350m stations (line 134), then why separate them out? Would it be better to include them with the <350 m stations because of similar response to the NAO?

We will revise the manuscript as suggested. The response of the non-Alpine stations is indeed similar to the NAO as for the continental stations: this is indicated by the sensitivity of the proxy to the NAO as well as by the calculated slopes. The slopes of the continental gradient in response to the NAO calculated for all continental stations (including the non-Alpine stations) are similar compared to one shown in the manuscript and are within the range of uncertainties.

Line 296: replace "confirmed" with "supported". From a semantics point of view, observations can "confirm," but models, being not real, can only support. We will revise the manuscript as suggested.

Line 337: change exceptions to exception, or provide another example. We will revise the manuscript as suggested.

Lines 411-417: the point about modern relationships maybe not being representative of past conditions is important and requires some more emphasis. We will revise the manuscript as suggested (see above).

Line 443: would read better as "situated in the Swiss Jura mountains approximately xxx km from the alpine divide. . ." We will revise the manuscript as suggested.

Line 446: equilibrium typo; line 446 "net" not "nett" We will revise the manuscript as suggested.

Line 457: There is no Figure 8 in the manuscript. Supplemental Figure? I would like to such a figure in the main text, as it is a crucial test of the current manuscript's hypothesis. We will revise the manuscript as suggested.

Line 458-460: this is where I would suggest the assumption of stationarity of the wNAOi and d18Opw is not supported. Certainly not for the "entire Holocene", but probably true for the past few millennia or so after ice sheets had decayed and land vegetation was established. One way around this problem is, for the pre-late Holocene, to refer to "NAO-like" behavior. We will revise the manuscript as suggested (see above).
* * *
[Figure]

**Fig. 1.** a) Correlation map between the wNAOi and the amount of precipitable water (PW) for the month December to March based on NCEP/NCER reanalysis data for the period 1948-2016 and the results of the longit

---

## Author Comment (AC2) · 1 Oct 2016

General Comments: It is not clear to me that the ECHAM5-wiso simulations add anything substantial to the manuscript. They seem to me to interfere with the flow and the communication of the main points of the research. The research does not provide a thorough test of the model, or a comparison with other existing models (both are beyond the scope of the manuscript). Because the manuscript uses measured data (e.g., NAO data, GNIP data), there exists little uncertainty regarding the quality of the data or the synoptic conditions associated with the data. So why include the model results,

which are considerably less certain than the actual data, and in fact are sometimes inconsistent with the other results, and presumably wrong? I would suggest either i) remove all mentions to the model or ii) move the model and associated discussion to the supplemental material. The strongest reason for keeping the model in is the independent analysis of the amount of precipitable water in the atmosphere - so maybe Figure S2 could be moved to the main text, with the 13 main GNIP sites marked on it, and the rest of the model discussion moved to the supplemental material. I think that a reference to the Supplemental Material would suffice as an explanation of how the new figure (the old S2) was constructed.

We will revise the manuscript as suggested. We will move the ECHAM5-wiso evaluation to the Supplementary Material and it will be discussed there in a new section and we will mention the ECHAM5-wiso results in the manuscript when it fits the discussion. We will additionally included NCEP/NCER reanalysis data of the amount of precipitable water to independently constrain the dependence of the amount of precipitable water on the wNAOi (see comment of Reviewer 1). The evaluations show that the dependence of the amount of precipitable water between the NCEP/NCER reanalysis data and the ECHAM5-wiso simulation are similar.

I wonder if a spatially stationary gradient can satisfactorily capture the complexity of the NAO. In particular, I suggest that the authors consider the 'Augmented NAO Index' of Wang et al., 2012, GRL within the context of their own work. The presumed stationarity of the pressure systems defining the NAO is an issue with most NAO studies, but because this study focuses on a spatial gradient perhaps it should focus more attention on this issue. Would the longitudinal gradient not change orientation through time as pressure centres shift location? Particularly in the Early Holocene, where more extensive sea ice would have relocated the pressure centers?

We will include the findings of Wang et al. (2012) in the discussion section of this manuscript and highlight potential caveats that are related to the spatial stationarity. If this would be the case we would expect to see that any reconstructed longitudinal

gradient based on speleothem $\delta$18O records has a positive slope (towards the east). However, this is not the case as McDermott et al. (2011) have demonstrated for the last 12 ka: instead the reconstruction of Holocene slopes reveals that the slopes of the longitudinal speleothem $\delta$18O gradients are steeper in the early Holocene and become progressively shallower until about 5 ka. This result demonstrates that if the pressure centres shift their location, as suggested from 9 ka to 8 ka by the recent study of Wassenburg et al. (2016) or throughout the Holocene as suggested by (Walczak et al., 2015), that this has only a minor influence on the orientation of the longitudinal gradient.

How dependent are the gradients calculated in Figure 4 on Valentia and Wallingford? Would the gradients be uniformly flat if these two sites were omitted? Even if this is the case, it does not imply a problem with the conclusions, though I suggest the authors investigate whether or not the gradients are driven by these two sites.

We will revise the manuscript as suggested. We will include an additional figure in the supplementary material. The results emphasise that the maritime stations have only a minor influence on temperature gradients. The effect is stronger for the absolute values of the precipitation gradients but the NAO dependence is only slightly modified. Therefore, conclusions drawn from the original dataset are not hampered when the maritime stations are included. We will include the results of this evaluation in the manuscript.

Also, should these be referred to as 'continental sites' or as 'maritime sites' particularly since they do seem to behave very differently than the other sites? We will revise the manuscript as suggested.

Specific Comments: Line 23: 'analyzed extensively' instead of 'with great effort', which suggests the analyses themselves were difficult. We will revise the manuscript as suggested.

Line 82: how was it determined that these stations had no Mediterranean influence?

[Figure]

We did not determine whether these stations have a physical influence of Mediterranean moisture, but labelled continental stations as non-Mediterranean influenced if they have a distance to the Mediterranean coastline >100km. We will include this information in the manuscript.

Line 447: 'nett' We will revise the manuscript as suggested.

Line 468: 'Central Europe' or 'central Europe' – be consistent We will revise the manuscript as suggested.

---

## Author Comment (AC3) · 1 Oct 2016

General and Specific Comments: Line 40 ff.: The second paragraph of the introduction is relatively long and mainly summarises what will be shown and discussed in the paper. This section could be shortened substantially to make the paper more concise. We will revise the manuscript as suggested if possible.

Line 46: "...to better evaluate the NAO-dependence on isotope longitudinal gradients." I thought the gradient depends on the state of the NAO and not the other way round? This should be clarified here and throughout the paper. We will revise the manuscript

as suggested.

Line 56: 37 stations have been analysed. 28 are GNIP stations, 6 are ANIP stations. What about the remaining 3? We will revise the manuscript as suggested. The remaining three stations were also GNIP stations.

Line 126 ff.: "Comparison of the longitudinal $\delta18O_{pw}$ and $\delta D_{pw}$ gradients derived from the ECHAM5-wiso with those from the station-based data show that slopes from the ECHAM5-wiso data reproduce the observed station-based slopes quite well (Figure 4)." I do not agree with this statement. All model data sets show a curve (i.e., the most negative slopes are shown for the 3rd and 4th NAO class) rather than a linear relationship with the NAO classes. It is, thus, misleading to state that the model data reproduce the station data "quite well". It would be good to see the fit statistics (slope, $r^2$ and p-value) not only for the station data, but for the model data as well (compare caption of Fig. 4). We will revise the manuscript as suggested. We will include the $r^2$ and p-values of the evaluation of the ECHAM5-wiso simulations in the figure caption of new Figure S3 (the ECHAM5-wiso evaluation is moved to the supplementary material, as suggested by Reviewer 2). The fit statistics show that the ECHAM5-wiso data has a weaker relationship the wNAOi. We will include these conclusions in the discussion of the ECHAM5-wiso results in the new section in the supplementary material.

Line 212 ff.: "Repeating the calculations using the vapour-ice phase change (snow) instead results in calculated differences that are still too small to explain the observed differences in $\delta18O_{pw}$ and $\delta D_{pw}$ between the western- and eastern-most stations (not shown)." Please provide a bit more information on this. I do not request a detailed discussion, but in the present form, the reader would not be able to do the calculations themselves if they wanted to. We will revise this sentence by clarifying what parameter is used and will state the values used. We will include the reference from which the values were used.

Line 217 ff.: "By contrast with the observational (GNIP) data discussed above, the

ECHAM5-wiso simulated differences in $\delta$18Opw and $\delta$Dpw can largely be accounted for by model air-temperature differences alone." ... "Overall, however, these results supports the conclusion above that the winter air temperature effect on the longitudinal winter $\delta$18Opw and $\delta$Dpw gradients is insufficient to explain the observed difference between the western and eastern GNIP stations." This is not clear to me and even appears contradictory. If the effects observed in the model data can be explained by the model temperatures, this does not support the conclusion derived from the data. Please clarify. In addition: "It is intriguing that the observed (GNIP datasets) and simulated (ECHAM5- wiso simulations) temperature slopes differ (Figure 6), while the slopes for longitudinal $\delta$18Opw and $\delta$Dpw gradients are apparently similar (Figure 4)." As far as I understood the text, the model temperatures in Eastern Europe are colder than those of the station data. If the d18O and dD values (i.e., the gradient and its dependence on the state of the NAO) are similar in the model and the data, but the temperatures are different, this either means that the dependence of model temperature on the NAO is too strong or that the sensitivity of the model d18O and dD values is too weak. In any case, this is an important difference, which makes it difficult to use the model data to interpret the station data. Based on the discussion following below, however, this statement is not necessary.

We will revised this section and deleted the sentence that is commented by the reviewer. The statement of this sentence is already given in the previous paragraph and it was confusing in the paragraph discussing the ECHAM5-wiso simulations. Note that the discussion on the ECHAM5-wiso data will be put in the supplementary material as suggested by Reviewer 2.

Line 330 ff.: "The reason for the different strength of these two mechanisms (temperature gradient and precipitation history) on the longitudinal $\delta$18Opw and $\delta$Dpw gradients for the observed (GNIP) and simulated (ECHAM5-wiso) datasets remains unclear, suggesting that the ECHAM5-wiso simulations warrant further investigation." I am not a climate modeller, but how representative are the precipitation data of the model for

the 13 stations (still a relatively low number) considered here. As far as I know, simulating (high-resolution) precipitation patterns is still difficult. Thus, the model data (which represent climate variability in a larger grid cell) may be more representative for the dependence of the west-east gradient in precipitation on the NAO than the station data. In summary, it is not very surprising for me that the precipitation data of the stations do not show a dependence on the state of the NAO, but the model data do. This is an interesting point that the reviewer highlights here. However, it is beyond the scope of this study to evaluate the robustness of the precipitation data here. This would require a rigorous comparison between different models and model types (GCM vs. regional models) and observational as well as reanalysis data. This should be and will be investigated in depth by a subsequent study that is developed at the moment.

Line 334 ff.: I would suggest to strongly shorten the first paragraph of section 4.2. It only summarises results from the analysis, which has partly already been presented in section 3.2. I would move all these results in section 3.2, and briefly summarise the findings here in one or two sentences. We will revise the manuscript as suggested.

Line 401 ff.: I would remove the reference to tree rings here, which mainly record summer climate (and water isotopes). We will revise the manuscript as suggested.

Line 426 ff.: "Note that our assumption about the temperature change represents a limiting case, because it implies that the annual air temperature, to which the cave air temperature is usually equilibrated, also decreases by the same value." As the authors state themselves, this assumption is not reasonable. It may be possible to find a correlation between winter and annual temperature (in the station and the model data). Based on that, one could try to estimate the dependence of annual temperature on the state of winter NAO. However, temperature is very stable in most caves, and an inter-annual change of 1.3 ậ̊ęC, as assumed in the example for the station Stuttgart, is almost impossible and may only occur in a strongly ventilated cave. In such caves, however, other effects, such as precipitation of $CaCO_3$ under conditions of disequilibrium stable isotope fractionation or evaporation, will probably dominate the d18O

values of the speleothem. The temperature effect may only be visible on a decadal or even longer time scale. Thus, the reference to persistent changes in the NAO on centennial to millennial time scales should be given at the beginning of the paragraph. However, I would rather suggest to remove the calculation because the caveats may not be present to many readers. We will revise the manuscript as suggested. We will clarify the calculations and state some more words on the caveats, like disequilibrium and smoothing effects to justify the points mentioned by the reviewer.

Section 5 in general: Since this section discusses the potential of speleothems for an NAO reconstruction based on speleothems, I miss a critical discussion of other potential "problems" of speleothems for NAO reconstruction (smoothing of the signals in the aquifer, contributions of different seasons than winter, disequilibrium stable isotope fractionation, dating uncertainties, etc.). I know that the authors are aware of these problems, so they should not be omitted from the discussion here. Mischel et al. (2015) have modelled some of these processes in detail. Their study could be referenced in this context. We will revise the manuscript as suggested. We will include a discussion on these caveats subsequently to the discussion of the NAO reconstruction. The study of Mischel et al. (2015) will be cited right at the beginning of the discussion of the NAO reconstructions.

Finally, as the two other reviewers question the stationarity of the NAO and the gradient in the past and, in particular, its dependence on the location of the pressure systems during the Early Holocene, it may be interesting to read and discuss the recent paper by Wassenburg et al. (2016). We will revise the manuscript as suggested. We will included the aforementioned study of Wassenburg et al. (2016) and extend the discussion on the stationarity of the NAO during the Holocene (see action on comments of reviewer 1 and 2).

---

## Author Response (AR1)

Actions on comments of reviewer 1-3 for the manuscript **'North Atlantic Oscillation controls on oxygen and hydrogen isotope gradients in winter precipitation across Europe; implications for palaeoclimate studies'**

Reviewer
Authors response and/or action

**Actions to the common comments of all reviewers**

All three reviewer commented on the stationarity of the NAO through the Holocene.

- **Reviewer 1 states:** I would suggest that the assumption of time-stationarity back to the early Holocene of the wNAOi and d18Opw is possibly correct but currently unsupported. The two sentences around Line 460 do not do this point justice. The manuscript would be stronger by either supporting the assertion with additional evidence that, say, the NAO existed more or less as we know it know during the early Holocene NAO, when the presence of upstream ice sheets and different insolation and vegetation regimes were present, or by providing stronger caveats for extending the NAO discussion prior to the late Holocene.

Revised. We have included recent studies that investigate the stationarity of the NAO during the Holocene and account their results in the discussion and interpretation of the findings of this study. Furthermore, we have extended the discussion on the stationarity of the NAO during the Holocene in general and highlight potential caveats.

Perhaps using the term "NAO-like" instead of the NAO, while emphasizing that early-Holocene climate had distinctly different forcings and boundary conditions than the late Holocene, would be advisable.

Revised.

- **Reviewer 2 states:** I wonder if a spatially stationary gradient can satisfactorily capture the complexity of the NAO. In particular, I suggest that the authors consider the 'Augmented NAO Index' of Wang et al., 2012, GRL within the context of their own work. The presumed stationarity of the pressure systems defining the NAO is an issue with most NAO studies, but because this study focuses on a spatial gradient perhaps it should focus more attention on this issue.

We have included the findings of (Wang et al., 2012) in the discussion section of this manuscript and highlight potential caveats that are related to the spatial stationarity.

**Would the longitudinal gradient not change orientation through time as pressure centres shift location?**

If this would be the case we would expect to see that any reconstructed longitudinal gradient based on speleothem $\delta^{18}O$ records has a positive slope (towards the east). However, this is not the case as McDermott et al. (2011) have demonstrated for the last 12 ka: instead the reconstruction of Holocene slopes reveals that the slopes of the longitudinal speleothem $\delta^{18}O$ gradients are steeper in the early Holocene and become progressively shallower until about 5 ka. This result demonstrates that if the pressure centres shift their location, as suggested from 9 ka to 8 ka by the recent study of Wassenburg et al. (2016), that this has only a minor influence on the orientation of the longitudinal gradient.

**Particularly in the Early Holocene, where more extensive sea ice would have relocated the pressure centers?**

Considering the results of McDermott et al. (2011) and Wassenburg et al. (2016) the shift of the location of the pressure centres is likely to be more gradual during the Holocene until stable environmental boundary conditions are reached. Taking account of the modelling results of Wassenburg et al. (2016) the retreat of the Laurentide ice sheet and a changing meltwater flux into the North Atlantic are responsible for the changes of the NAO (or as stated in Wassenburg et al. (2016) the 'reorganisation' of the NAO) and the shift of the pressure systems particularly of the Icelandic low: during the course of the Holocene the simulations would suggest that the Icelandic low system is shifted towards north-east.

- **Reviewer 3 states:** Finally, as the two other reviewers question the stationarity of the NAO and the gradient in the past and, in particular, its dependence on the location of the pressure systems during

the Early Holocene, it may be interesting to read and discuss the recent paper by Wassenburg et al. (2016).

*Revised. We have included the aforementioned study of Wassenburg et al. (2016) and have extended the discussion on the stationarity of the NAO during the Holocene.*

**Actions to the comments of Reviewer 1**

*General Comments:*
I would suggest that the assumption of time-stationarity back to the early Holocene of the wNAOi and d18Opw is possibly correct but currently unsupported. The two sentences around Line 460 do not do this point justice. The manuscript would be stronger by either supporting the assertion with additional evidence that, say, the NAO existed more or less as we know it know during the early Holocene NAO, when the presence of upstream ice sheets and different insolation and vegetation regimes were present, or by providing stronger caveats for extending the NAO discussion prior to the late Holocene. Perhaps using the term "NAO-like" instead of the NAO, while emphasizing that early-Holocene climate had distinctly different forcings and boundary conditions than the late Holocene, would be advisable.

*Revised. This is an issue raised by all reviewers and its detailed comment can be found at the beginning of this document.*

One other conclusion is pretty easily testable but doesn't seem to have been evaluated rigorously: that precipitable water is less during negative wNAOi states. The current study would have much stronger standing with readers if estimates of precipitable water (say from the NCEP database) and wNAOi were compared directly.

*Revised. We have included NCEP/NCER reanalysis data of precipitable water. We evaluated the dependence of the precipitable water on the wNAOi over Europe and have added a new figure in the manuscript.*

Specific Comments:
Line 122: change to "more strongly negative".
*Revised.*
Please give the altitudes of the >350m non-alpine stations. If they don't differ much from the <350m stations (line 134), then why separate them out? Would it be better to include them with the <350 m stations because of similar response to the NAO?
*Revised. The response of the non-Alpine stations is indeed similar to the NAO as for the continental stations: this is indicated by the sensitivity of the proxy to the NAO as well as by the calculated slopes. The slopes of the continental gradient in response to the NAO calculated for all continental stations (including the non-Alpine stations) are similar compared to one shown in the manuscript and are within the range of uncertainties.*
Line 296: replace "confirmed" with "supported". From a semantics point of view, ob- servations can "confirm," but models, being not real, can only support.
*Revised.*
Line 337: change exceptions to exception, or provide another example.
*Revised.*
Lines 411-417: the point about modern relationships maybe not being representative of past conditions is important and requires some more emphasis.
*Revised (see revision for general comments).*
Line 443: would read better as "situated in the Swiss Jura mountains approximately xxx km from the alpine divide. . ."
*Revised.*
Line 446: equilibrium typo; line 446 "net" not "nett"
*Revised.*
Line 457: There is no Figure 8 in the manuscript. Supplemental Figure? I would like to such a figure in the main text, as it is a crucial test of the current manuscript's hypothesis.
*Revised.*

Line 458-460: this is where I would suggest the assumption of stationarity of the wNAOi and d18Opw is not supported. Certainly not for the "entire Holocene", but probably true for the past few millennia or so after ice sheets had decayed and land vegetation was established. One way around this problem is, for the pre-late Holocene, to refer to "NAO-like" behavior.

Revised. See also comment in the Section General Comments.

**Actions to the comments of Reviewer 2**

*General Comments:*

It is not clear to me that the ECHAM5-wiso simulations add anything substantial to the manuscript. They seem to me to interfere with the flow and the communication of the main points of the research. The research does not provide a thorough test of the model, or a comparison with other existing models (both are be- yond the scope of the manuscript). Because the manuscript uses measured data (e.g., NAO data, GNIP data), there exists little uncertainty regarding the quality of the data or the synoptic conditions associated with the data. **So why include the model results, which are considerably less certain than the actual data, and in fact are sometimes inconsistent with the other results, and presumably wrong? I would suggest either i) remove all mentions to the model or ii) move the model and associated discussion to the supplemental material.** The strongest reason for keeping the model in is the independent analysis of the amount of precipitable water in the atmosphere - so maybe Figure S2 could be moved to the main text, with the 13 main GNIP sites marked on it, and the rest of the model discussion moved to the supplemental material. I think that a reference to the Supplemental Material would suffice as an explanation of how the new figure (the old S2) was constructed.

Revised. We have moved the ECHAM5-wiso evaluation to the Supplementary Material and it is discussed there in a new section and mention the results in the manuscript when it fits the discussion. We have additionally included NCEP/NCER reanalysis data of the amount of precipitable water to independently constrain the dependence of the amount of precipitable water on the wNAOi. The evaluations show that the dependence of the amount of precipitable water between the NCEP/NCER reanalysis data and the ECHAM5-wiso simulation are similar.

I wonder if a spatially stationary gradient can satisfactorily capture the complexity of the NAO. In particular, I suggest that the authors consider the 'Augmented NAO Index' of Wang et al., 2012, GRL within the context of their own work. The presumed stationarity of the pressure systems defining the NAO is an issue with most NAO studies, but because this study focuses on a spatial gradient perhaps it should focus more attention on this issue. Would the longitudinal gradient not change orientation through time as pressure centres shift location? Particularly in the Early Holocene, where more extensive sea ice would have relocated the pressure centers?

Revised. This is an issue raised by all reviewers and its detailed comment can be found at the beginning of this document.

How dependent are the gradients calculated in Figure 4 on Valentia and Wallingford? Would the gradients be uniformly flat if these two sites were omitted? Even if this is the case, it does not imply a problem with the conclusions, though I suggest the authors investigate whether or not the gradients are driven by these two sites.

Revised. We have included an additional figure in the supplementary material. The results emphasise that the maritime stations have only a minor influence on temperature gradients. The effect is stronger for the absolute values of the precipitation gradients but the NAO dependence is only slightly modified. Therefore, conclusions drawn from the original dataset are not hampered by the including the maritime stations.

Also, should these be referred to as 'continental sites' or as 'maritime sites' particularly since they do seem to behave very differently than the other sites?

Revised.

Specific Comments:

Line 23: 'analyzed extensively' instead of 'with great effort', which suggests the analyses themselves were difficult.

Revised.

Line 82: how was it determined that these stations had no Mediterranean influence?

We did not determine whether these stations have a physical influence of Mediterranean moisture, but labelled continental stations as non-Mediterranean influenced if they have a distance to the Mediterranean coastline >100km. We have included this information in the manuscript.

Line 447: 'nett'

Revised.

Line 468: 'Central Europe' or 'central Europe' – be consistent

Revised.

**Actions to the comments of Reviewer 3**

*General and Specific Comments:*

Line 40 ff.: The second paragraph of the introduction is relatively long and mainly summarises what will be shown and discussed in the paper. This section could be shortened substantially to make the paper more concise.

Revised. We shortened it slightly.

Line 46: "...to better evaluate the NAO-dependence on isotope longitudinal gradients." I thought the gradient depends on the state of the NAO and not the other way round? This should be clarified here and throughout the paper.

Revised.

Line 56: 37 stations have been analysed. 28 are GNIP stations, 6 are ANIP stations. What about the remaining 3?

Revised.

Line 126 ff.: "Comparison of the longitudinal $\delta18O_{pw}$ and $\delta D_{pw}$ gradients derived from the ECHAM5-wiso with those from the station-based data show that slopes from the ECHAM5-wiso data reproduce the observed station-based slopes quite well (Figure 4)." I do not agree with this statement. All model data sets show a curve (i.e., the most negative slopes are shown for the 3rd and 4th NAO class) rather than a linear relationship with the NAO classes. It is, thus, misleading to state that the model data reproduce the station data "quite well". **It would be good to see the fit statistics (slope, r2 and p-value) not only for the station data, but for the model data as well (compare caption of Fig. 4).**

Revised. We have included the $r^2$ and p-values of the evaluation of the ECHAM5-wiso simulations in the figure caption of Figure S3.

Line 212 ff.: "Repeating the calculations using the vapour-ice phase change (snow) instead results in calculated differences that are still too small to explain the observed differences in $\delta18O_{pw}$ and $\delta D_{pw}$ between the western- and eastern-most stations (not shown)." Please provide a bit more information on this. I do not request a detailed discussion, but in the present form, the reader would not be able to do the calculations themselves if they wanted to.

We have revised this sentence by clarifying what parameter is used, stated its value that reader can repeat the calculation and have included the reference from which the values were used.

Line 217 ff.: "By contrast with the observational (GNIP) data discussed above, the ECHAM5-wiso simulated differences in $\delta18O_{pw}$ and $\delta D_{pw}$ can largely be accounted for by model air-temperature differences alone." ... "Overall, however, these results supports the conclusion above that the winter air temperature effect on the longitudinal winter $\delta18O_{pw}$ and $\delta D_{pw}$ gradients is insufficient to explain the observed difference between the western and eastern GNIP stations." This is not clear to me and even appears contradictory. If the effects observed in the model data can be explained by the model temperatures, this does not support the conclusion derived from the data. Please clarify.

In addition: "It is intriguing that the observed (GNIP datasets) and simulated (ECHAM5- wiso simulations) temperature slopes differ (Figure 6), while the slopes for longitudinal δ18Opw and δDpw gradients are apparently similar (Figure 4)." As far as I understood the text, the model temperatures in Eastern Europe are colder than those of the station data. If the d18O and dD values (i.e., the gradient and its dependence on the state of the NAO) are similar in the model and the data, but the temperatures are different, this either means that the dependence of model temperature on the NAO is too strong or that the sensitivity of the model d18O and dD values is too weak. In any case, this is an important difference, which makes it difficult to use the model data to interpret the station data. Based on the discussion following below, however, this statement is not necessary.

We have revised this section and deleted the sentence that is commented by the reviewer. The statement of this sentence is already given in the previous paragraph and it was confusing in the paragraph discussing the ECHAM5-wiso simulations.

Line 330 ff.: "The reason for the different strength of these two mechanisms (temperature gradient and precipitation history) on the longitudinal δ18Opw and δDpw gradients for the observed (GNIP) and simulated (ECHAM5-wiso) datasets remains unclear, suggesting that the ECHAM5-wiso simulations warrant further investigation." I am not a climate modeller, but how representative are the precipitation data of the model for the 13 stations (still a relatively low number) considered here. As far as I know, simulating (high-resolution) precipitation patterns is still difficult. **Thus, the model data (which represent climate variability in a larger grid cell) may be more representative for the dependence of the west-east gradient in precipitation on the NAO than the station data. In summary, it is not very surprising for me that the precipitation data of the stations do not show a dependence on the state of the NAO, but the model data do.**

This is an interesting point that the reviewer highlights here. However, it is beyond the scope of this study to evaluate the robustness of the precipitation here. This would require a rigorous comparison between different models and model types (GCM vs. regional models) and observational as well as reanalysis data.

Line 334 ff.: I would suggest to strongly shorten the first paragraph of section 4.2. It only summarises results from the analysis, which has partly already been presented in section 3.2. I would move all these results in section 3.2, and briefly summarise the findings here in one or two sentences.

Revised.

Line 401 ff.: I would remove the reference to tree rings here, which mainly record summer climate (and water isotopes).

Revised.

Line 426 ff.: "Note that our assumption about the temperature change represents a limiting case, because it implies that the annual air temperature, to which the cave air temperature is usually equilibrated, also decreases by the same value." As the authors state themselves, this assumption is not reasonable. It may be possible to find a correlation between winter and annual temperature (in the station and the model data). Based on that, one could try to estimate the dependence of annual temperature on the state of winter NAO. However, temperature is very stable in most caves, and an inter-annual change of 1.3 ∘C, as assumed in the example for the station Stuttgart, is almost impossible and may only occur in a strongly ventilated cave. In such caves, however, other effects, such as precipitation of CaCO3 under conditions of disequilibrium stable isotope fractionation or evaporation, will probably dominate the d18O values of the speleothem. The temperature effect may only be visible on a decadal or even longer time scale. **Thus, the reference to persistent changes in the NAO on centennial to millennial time scales should be given at the beginning of the paragraph.** However, I would rather suggest to remove the calculation because the caveats may not be present to many readers.

Revised. We clarified the calculations and state some more words on the caveats.

Section 5 in general: Since this section discusses the potential of speleothems for an NAO reconstruction based on speleothems, **I miss a critical discussion of other potential "problems" of speleothems for NAO reconstruction (smoothing of the signals in the aquifer, contributions of**

**different seasons than winter, disequilibrium stable isotope fractionation, dating uncertainties, etc.). I know that the authors are aware of these problems, so they should not be omitted from the discussion here. Mischel et al. (2015) have modelled some of these processes in detail. Their study could be referenced in this context.**

Revised. We have included a discussion on these caveats subsequently to the discussion of the NAO reconstruction. The study of Mischel et al. (2015) is cited right at the beginning of the discussion of the NAO recpnstructions.

Finally, as the two other reviewers question the stationarity of the NAO and the gradient in the past and, in particular, its dependence on the location of the pressure systems during the Early Holocene, it may be interesting to read and discuss the recent paper by Wassenburg et al. (2016).

Revised. This is an issue raised by all reviewers and its detailed comment can be found at the beginning of this document.

[revised manuscript text omitted]